# Liver ACOX1 regulates levels of circulating lipids that promote metabolic health through adipose remodeling

Dongliang Lu[1], Anyuan He[1,4], Min Tan[1], Marguerite Mrad [1], Amal El Daibani[2], Donghua Hu[1], Xuejing Liu[1], Brian Kleiboeker [1], Tao Che [2], Fong-Fu Hsu[1], Monika Bambouskova [1], Clay F. Semenkovich [1,3] & Irfan J. Lodhi [1] ✉

The liver gene expression of the peroxisomal β-oxidation enzyme acyl-coenzyme A oxidase 1 (ACOX1), which catabolizes very long chain fatty acids (VLCFA), increases in the context of obesity, but how this pathway impacts systemic energy metabolism remains unknown. Here, we show that hepatic ACOX1-mediated β-oxidation regulates inter-organ communication involved in metabolic homeostasis. Liver-specific knockout of *Acox1* (*Acox1*-LKO) protects mice from diet-induced obesity, adipose tissue inflammation, and systemic insulin resistance. Serum from *Acox1*-LKO mice promotes browning in cultured white adipocytes. Global serum lipidomics show increased circulating levels of several species of ω−3 VLCFAs (C24-C28) with previously uncharacterized physiological role that promote browning, mitochondrial biogenesis and Glut4 translocation through activation of the lipid sensor GPR120 in adipocytes. This work identifies hepatic peroxisomal β-oxidation as an important regulator of metabolic homeostasis and suggests that manipulation of ACOX1 or its substrates may treat obesity-associated metabolic disorders.

Obesity is a serious public health problem associated with a wide range of comorbidities, including type 2 diabetes, cardiovascular disease, and hepatic steatosis. Dysfunctional lipid metabolism is a hallmark of obesity and is implicated in the pathogenesis of these comorbidities, with an imbalance between lipid synthesis and degradation contributing to the accumulation of excess fat in adipose tissue and other organs[1,2]. Therapeutic expansion or activation of brown adipose tissue (BAT) or the related beige fat, which is an inducible form of adipose cells that emerge in subcutaneous white adipose tissue (WAT) in response to certain external stimuli, is a promising strategy to treat obesity and its complications. Brown adipocytes and beige adipocytes possess high mitochondrial content and mediate thermogenesis through multiple mechanisms, most notably through a process involving uncoupling protein-1 (UCP1), a mitochondrial protein that

disassociates respiration from ATP synthesis to generate heat. Activation of thermogenic fat promotes resting energy expenditure, whole-body glucose disposal, and insulin sensitivity[3].

The liver is also an important regulator of metabolic homeostasis. It acts as a hub for lipid metabolism, coordinating the transport, synthesis, storage, and degradation of lipids in response to changes in nutrient availability and energy demand. Fatty acid oxidation, the process by which fatty acids are broken down to produce energy, is one of the major pathways involved in lipid degradation, with the liver playing a critical role in this process.

Fatty acids can be oxidized in both mitochondria and peroxisomes. While the two pathways share some similarities, they differ in their substrate specificity. Mitochondrial β-oxidation plays a more important role in the catabolism of short-, medium-, and most long-

[1]Division of Endocrinology, Metabolism & Lipid Research, Washington University School of Medicine, St. Louis, MO 63110, USA. [2]Center for Clinical Pharmacology, Department of Anesthesiology, Washington University School of Medicine, St. Louis, MO 63110, USA. [3]Department of Cell Biology and Physiology; Washington University School of Medicine, St. Louis, MO 63110, USA. [4]Present address: School of Life Sciences, Anhui Medical University, Hefei 230032, China. ✉e-mail: ilodhi@wustl.edu

chain fatty acids, while peroxisomal β-oxidation is required for the degradation of very-long-chain fatty acids (VLCFAs; ≥22 carbons), di- and trihydroxycholestanoic acid (DHCA and THCA), long chain dicarboxylic acids, 2-methyl branched-chain fatty acids (BCFA), leukotrienes and prostaglandins[4,5]. The first and rate-limiting step of peroxisomal ß -oxidation is carried out by the acyl-coenzyme A oxidase family of proteins, which consists of three members. ACOX1 is specific to the degradation of straight-chain saturated and unsaturated VLCFAs. ACOX2 promotes β-oxidation of 2-methyl-BCFA, such as pristanic acid, and bile acid intermediates, including DHCA and THCA. ACOX3 is also involved in catabolism of 2-methyl branched fatty acids, suggesting that it has a partially overlapping function with ACOX2[6].

Through its role in the oxidation of certain polyunsaturated fatty acids, ACOX1-mediated β-oxidation affects the levels of the omega-3 (ω−3) fatty acids eicosapentaenoic acid (EPA; C20:5) and docosahexaenoic acid (DHA; C22:6). In mammals, peroxisomal β-oxidation not only oxidizes DHA, but also affects the production of DHA through a process involving desaturation and elongation of α-linolenic acid (ALA, 18:3-3) to generate tetracosahexaenoic acid (THA; C24:6), which is then chain-shortened to DHA by peroxisomal β-oxidation[7,8]. Peroxisome β-oxidation deficiency results in decreased levels of DHA and accumulation of longer chain fatty acids[9].

ACOX1 is enriched in the liver. Acox1[-/-] mice have severely impaired VLCFA β-oxidation in the liver[10]. Our previous studies show that mice with liver-specific knockout of ACOX1 appear normal and are protected against hepatic steatosis through induction of lipophagy, autophagic degradation of lipid droplets. Mechanistically, acetyl-CoA derived from ACOX1-mediated β-oxidation promotes acetylation of Raptor, a component of the mTORC1 complex which inhibits lipophagy[11]. However, the impact of ACOX1-mediated β-oxidation in the liver on whole-body energy homeostasis, especially in the context of diet-induced obesity, has not been investigated.

Here, we report that liver-specific knockout of ACOX1 promotes resistance to diet-induced obesity, inflammation, and insulin resistance. Mechanistically, disruption of hepatic peroxisomal β-oxidation leads to increased circulating levels of ω-3 VLCFA with previously uncharacterized physiological functions, especially C24:6n-3 and C26:6n-3. Our results show that these polyunsaturated fatty acids activate the G-protein coupled receptor GPR120, promoting browning of white adipocytes and increasing insulin-stimulated Glut4 translocation in adipocytes.

## Results

### Hepatic ACOX1 deficiency protects against HFD-induced obesity, insulin resistance, and inflammation

Peroxisomes are involved in the β-oxidation of VLCFA (Fig. 1a). Very long-chain fatty acyl-CoA are transported into peroxisomes for β-oxidation by ABCD1 or ABCD2, which have distinct substrate specificities[12]. The first and rate-limiting step of VLCFA β-oxidation is catabolized by ACOX1, which is enriched in the liver[5]. The hepatic gene expression of *Acox1*, as well as other enzymes in the pathway, increased with high-fat feeding in mice (Fig. 1b). Search of publicly available transcriptome profiling data (GEO: GSE61260 comparing liver gene expression of 58 lean control and 76 individuals with obesity, revealed that *Acox1* gene expression level is significantly higher in the livers of individuals with obesity than in lean controls (Supplementary Fig. 1a). Moreover, the protein levels of the peroxisomal biogenesis factors PEX5, PEX16, and PMP70 were increased in the livers of high fat diet-fed mice (Supplementary Fig. 1b), suggesting that liver peroxisomes regulate diet-induced obesity and related complications. To study the role of hepatic peroxisomal β-oxidation in regulating whole-body energy metabolism, we studied mice with liver-specific knockout of *Acox1* (*Acox1*-LKO mice). Although there was no difference in body weight between the genotypes in animals fed normal chow diet (Supplementary Fig. 1c), high-fat diet-fed *Acox1*-LKO male mice

weighed significantly less (Fig. 1c). Body composition analysis showed that the knockout mice had decreased fat mass (Fig. 1d). Subcutaneous fat, including inguinal white adipose tissue (iWAT), weighed significantly less in HFD-fed *Acox1*-LKO mice, but gonadal WAT (gWAT), BAT and liver weights were not affected (Fig. 1e). Figure 1f shows a gross image of iWAT depots from *Acox1*-LKO and control mice. Histologic analysis indicated that adipocytes in iWAT were smaller (Fig. 1g). Quantification of cell size showed that adipocytes in iWAT were significantly smaller in *Acox1*-LKO mice (Fig. 1h).

We next assessed the effect of hepatic *Acox1* deletion on systemic glucose homeostasis. HFD-fed *Acox1*-LKO male mice exhibited improved glucose tolerance (Fig. 1i) in the context of nearly significantly decreased serum insulin levels ($p = 0.059$) (Fig. 1j) and significantly decreased serum triacylglycerol (TAG) content (Fig. 1k), indicating that hepatic ACOX1 deficiency promotes insulin sensitivity. Insulin tolerance testing confirmed the increased insulin sensitivity, though the results were confounded by decreased basal glucose levels in the KO mice (Supplementary Fig. 1d). Blood glucose excursion following pyruvate injection was almost significantly ($p = 0.053$) decreased in *Acox1*-LKO mice (Supplementary Fig. 1e), likely reflecting an improved ability of insulin to inhibit hepatic gluconeogenesis. Western blot analysis of insulin-stimulated Akt (Ser473) phosphorylation in the liver (Fig. 1l) as well as gWAT and iWAT (Fig. 1m) further confirmed improved insulin signaling in *Acox1*-LKO mice. Notably, HFD-fed *Acox1*-LKO female mice had no difference in body weight compared to control animals (Supplementary Fig. 1f) but exhibited significantly improved systemic glucose homeostasis (Supplementary Fig. 1g), suggesting that hepatic ACOX1 inactivation has a primary effect on insulin sensitivity, independent of body weight difference.

Obesity is associated with adipose tissue inflammation, which contributes to co-morbidities, including insulin resistance and non-alcoholic fatty liver disease (NAFLD)[13]. Adipose tissue inflammation in the context of obesity is more intense in visceral WAT as compared to subcutaneous WAT[14,15]. To determine if HFD-fed *Acox1*-LKO mice are protected against obesity-associated infiltration of immune cells in adipose tissue, we used high-parameter spectral flow cytometry to quantify leukocytes (cells positive for common leukocyte antigen CD45) in stromal vascular fraction (SVF) cells from gWAT, a visceral adipose depot (Supplementary Fig. 1h). Despite the weight of gWAT depots in *Acox1*-LKO mice trending higher (Supplementary Fig. 1i), the number of SVF cells and CD45+ cells per gram of fat depot was lower in *Acox1*-LKO mice, demonstrating decreased immune infiltration (Supplementary Fig. 1j, k). Using unbiased clustering analysis of CD45+ cells, we have identified 12 distinct immune cell types, among them adipose tissue macrophages (ATMs) (Fig. 1n). Notably, this analysis revealed a CD9 + ATM subset, recognized for its association with pathological obesity[16] (Supplementary Fig. 1l). Quantification of per gram cell numbers showed normal number of various immune cells (Supplementary Fig. 1m−o), except ATMs, which were significantly decreased in *Acox1*-LKO (Fig. 1o), and this decrease was mostly driven by lowered numbers of CD9+ ATMs subset in *Acox1*-LKO (Fig. 1p). Cell density plots suggested phenotypic remodeling in the ATM population in *Acox1*-LKO (Fig. 1q). This was accompanied by increased proportions of CD11c-CD206+ macrophages that are known to represent anti-inflammatory phenotype[17,18] (Fig. 1r). Simultaneously, there was a decrease in pro-inflammatory macrophages CD11c + CD206+ ATMs which are associated with obesity and insulin resistance[19] (Fig. 1s). These data suggest decreased adipose tissue inflammatory status in *Acox1*-LKO as indicated by reduced infiltration by inflammatory ATM subsets, consistent with the improved metabolic profile. In further support of the decreased presence of pro-inflammatory ATM in the mutant mice, the infiltration of F4/80-positive macrophages was decreased in gWAT of HFD-fed *Acox1*-LKO mice (Fig. 1t), a result corroborated by significantly decreased mRNA levels of F4/80 and IL6 (Supplementary Fig. 1p), a marker of chronic inflammation. Together,

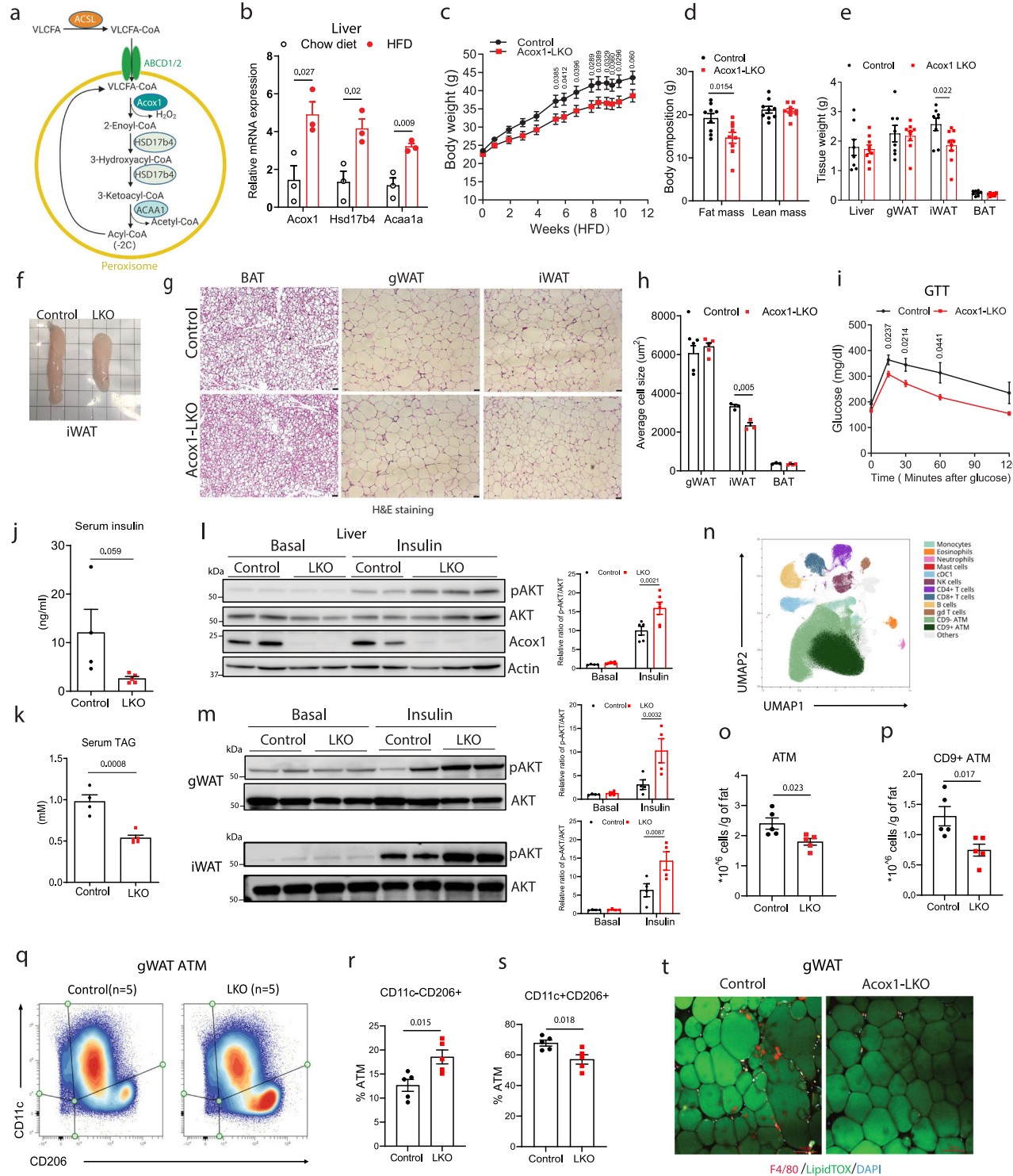

these results indicate that liver-specific inactivation of peroxisomal β-oxidation promotes resistance to diet-induced obesity and inflammation and promotes insulin sensitivity.

## Acox1-LKO mice exhibit adipose tissue browning and increased energy expenditure

The decreased adiposity resulting from hepatic *Acox1* inactivation was not associated with altered food intake (Supplementary Fig. 2a), ambulatory movement (Supplementary Fig. 2b), or intestinal absorption of fat (Supplementary Fig. 2c). Instead, indirect calorimetry revealed that oxygen consumption (VO$_2$) (Fig. 2a) and carbon dioxide

emission (VCO$_2$) (Supplementary Fig. 2d) were elevated. Regression-based analysis of energy expenditure confirmed that hepatic ACOX1 deficiency results in increased basal energy expenditure (Fig. 2b). Respiratory exchange ratio (RER) was not significantly altered (Supplementary Fig. 2e).

The elevated resting energy expenditure prompted us to determine whether *Acox1*-LKO mice have browning of white fat. Gene expression analysis demonstrated that brown adipocyte genes, including *Ucp1*, *Elovl3,* and *Cidea* were upregulated in iWAT of chow or HFD-fed male and female mice (Fig. 2c and Supplementary Fig. 2f, g). The magnitude of the increase was greater in chow-fed animals,

**Fig. 1 | Liver-specific knockout of *Acox1* protects against high-fat diet-induced obesity, inflammation, and insulin resistance in mice. a** Peroxisomal VLCFA β-oxidation pathway. **b** Peroxisomal fatty acid β-oxidation gene expression in liver of mice fed with chow or HFD for 16 weeks (*n* = 3). **c** Body weight of control and *Acox1*-LKO male mice fed with HFD (*n* = 9). **d** MRI analysis of mice after HFD feeding (*n* = 9). **e** Tissue weights of HFD-fed mice; *n* = 8 (control), *n* = 9 (*Acox1*-LKO). **f** Representative gross images of iWAT from HFD-fed control and *Acox1*-LKO mice. **g** H&E staining of adipose tissues from control and *Acox1*-LKO mice (scale bar, 50 μm). **h** Cell size quantification in BAT (*n* = 3 for control, 4 for Acox1-LKO), gWAT (*n* = 5) and iWAT (*n* = 3). **i** Glucose tolerance test of HFD-fed control (*n* = 9) and *Acox1*-LKO (*n* = 11) mice. **j–k** Serum insulin level (**j**) and TAG level (**k**) of control (*n* = 4) and *Acox1*-LKO (*n* = 5) mice. **l** Akt phosphorylation in livers of mice at baseline and 10 min after insulin injection. Quantification based on *n* = 4 (basal), 5 (control with insulin) and 6 (LKO with insulin) mice. **m** Akt phosphorylation in

gWAT and iWAT of mice at baseline and 10 min after insulin injection. Quantification based on *n* = 4/group. **n** UMAP visualization of CD45+ cells in mice gWAT; ATM, adipose tissue macrophages. **o–p** Number of macrophages (**o**) and CD9 + ATM (**p**) per gram of fat (*n* = 5). **q** Density plots of macrophages subpopulation expressing CD11c or CD206 (*n* = 5). **r–s** Percentage of CD11c-CD206+ (**r**) and CD11c + CD206+ (**s**) in total gWAT macrophages (*n* = 5). **t** F4/80 (red) staining in gWAT whole tissue mounts stained with LipidTOX (green) (scale bar, 100 μm). Data in (**b–e, h–m, o–p, and r–s**) are from biologically independent samples. Images in (**f**) and (**g**) are representative of three mice/genotype. Images in (**t**) are representative of two separate experiments. Data with error bars are reported as the mean ± SEM. *P* values were determined by two-sided unpaired Student's *t* test in (**b, d–e, h, j, k, o–p, r, s**) or two-way ANOVA followed by Fisher's LSD test in **c, i** and **l-m**. Source data are provided as a Source Data file.

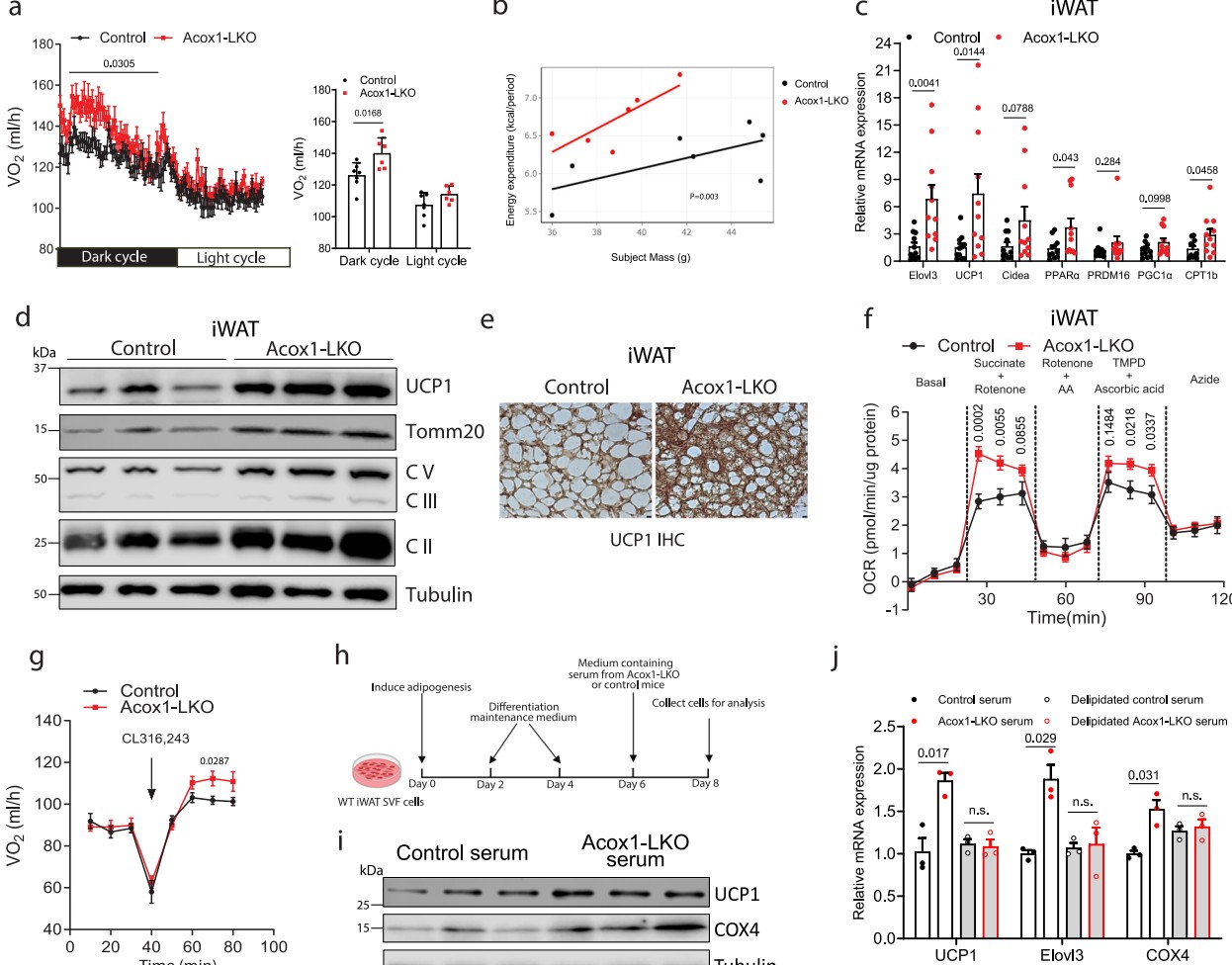

**Fig. 2 | Hepatic ACOX1 deficiency promotes basal energy expenditure and browning of subcutaneous white adipose tissue. a** Oxygen consumption rat (VO₂) of HFD-fed control (*n* = 7) and *Acox1*-LKO (*n* = 6) mice. **b** Analysis of energy expenditure in control and *Acox1*-LKO mice. **c** Browning gene expression in iWAT of control and *Acox1*-LKO mice fed normal chow diet (*n* = 11). **d** Western blot analysis of UCP1, Tomm20 and mitochondrial respiratory complex V, III, and II in iWAT from mice after a 2-day cold exposure (*n* = 3). **e** UCP1 immunohistochemistry in iWAT from control and *Acox1*-LKO mice after cold exposure (*n* = 2; scale bar, 25 μm). **f** Activities of mitochondrial complex II and complex IV from mice iWAT were measured using Seahorse (*n* = 10). **g** VO₂ of control and *Acox1*-LKO mice before and

after CL316,243 injection (*n* = 7). **h** Strategy for treatment of adipocytes derived from WT iWAT with serum from *Acox1*-LKO or control mice. Created with BioRender.com. **i** Western bolt analysis of UCP1 and COX4 (*n* = 3). **j** Browning gene expression in iWAT adipocytes treated with whole or delipidated mouse serum (*n* = 3). Data in (**a–d, f, g, i, and j**) are from biologically independent samples. Images in (**e**) are representative of two separate experiments. Data with error bars are the mean ± SEM. *P* values were determined by two-sided unpaired Student's *t* test in (**a**) (right panel) and (**c**), two-way ANOVA with Tukey's honest difference post hoc test in (**a**) (left panel), ANCOVA in (**b**), two-way ANOVA followed by Fisher's LSD test in (**f, g,** and **j**). n.s., not significant. Source data are provided as a Source Data file.

presumably because high-fat feeding dramatically decreases thermogenic gene expression in iWAT, as previously reported[20]. The expression of thermogenics genes was not significantly altered in classical BAT (Supplementary Fig. 2h). The increased expression of UCP1 and mitochondrial proteins in iWAT was confirmed by Western blot analysis and immunohistochemistry (Fig. 2d, e and Supplementary Fig. 2i). Moreover, Complex II- and Complex IV- dependent respiration activities were significantly higher in iWAT of Acox1-LKO mice (Fig. 2f). Indirect calorimetry revealed that β3-adrenoreceptor-dependent $O_2$ consumption ($VO_2$), a proxy for thermogenic fat activation, was significantly increased in Acox1-LKO mice (Fig. 2g). Together, these results implicate liver-to-adipose tissue signaling in adipocyte browning and increased energy expenditure.

To investigate a potential role for inter-organ communication in adipocyte browning, we isolated SVF cells from iWAT of wild-type mice, differentiated these cells into adipocytes and then treated them with serum from Acox1-LKO or control mice (Fig. 2h). Treatment of white adipocytes with serum from Acox1-LKO mice increased UCP1 and COX4 protein levels (Fig. 2i) and increased thermogenic gene expression, while this effect was not observed when the cells were treated with delipidated serum (Fig. 2j). These results suggest that hepatic ACOX1 deficiency promotes adipocyte browning by altering levels of circulating signaling lipid(s).

## Polyunsaturated ω-3 very long chain fatty acids accumulate in serum of Acox1-LKO mice

To understand potential mechanisms through which genetic inactivation of the liver Acox1 might affect adipose tissue browning, we used a combination of liver transcriptomics and untargeted serum lipidomics (Fig. 3). RNA-seq analysis was performed to identify gene expression changes induced by Acox1 knockout in the liver. KEGG enrichment analysis indicated that altered genes represent several lipid metabolism pathways (Fig. 3a). Of the 63 altered lipid metabolism-related genes, most were upregulated (Fig. 3b). Further analysis revealed that these genes are involved in various lipid metabolism processes, including fatty acid transport, activation, elongation and desaturation (Fig. 3c). In addition, the disruption of peroxisomal β-oxidation was associated with compensatory increases in mitochondrial β-oxidation and ER-localized ω-oxidation (Fig. 3c), a relatively minor pathway under normal physiological circumstances that results in production of dicarboxylic acids.

To determine if these gene expression changes translate to alterations in liver and circulating lipids, we profiled the liver and serum lipidome. Gene expression changes for ω-oxidation pathway were notable given that multiple genes encoding proteins in the first and third steps were increased, while the second step genes were decreased (Supplementary Fig. 3a). Since the first step yields hydroxy fatty acids, such as 20-HETE[21], we measured these fatty acids in Acox1-LKO mice and mass spectrometric analysis indicated that liver and serum hydroxy fatty acids were unaffected (Supplementary Fig. 3b, c).

To assay levels of a broader array of lipid species, we performed untargeted lipidomics on serum from control and Acox1-LKO mice (Fig. 3d). Levels of many lipids were up or down regulated in the serum of mice with hepatic Acox1 knockout. Relationship between liver transcriptomic and serum lipidomic changes is illustrated using a Circos plot (Fig. 3e). Further analysis of the lipidomics data using MetaboAnalyst[22] identified pathway enrichment for biosynthesis of unsaturated fatty acids among serum lipids from Acox1-LKO mice (Fig. 3f). Of note, the circulating levels of several species of ω-3 VLCFA, including C24:6 (tetracosahexaenoic acid; THA), C26:6 (hexacosahexaenoic acid; HHA), and C28:6 (octacosahexaenoic acid; OHA) significantly increased in Acox1-LKO mice (Fig. 3g). In contrast to the increases in VLCFA, levels of EPA and DHA, two common ω-3 fatty acids, were significantly decreased in Acox1-LKO mice (Fig. 3g). To validate these results and quantify total concentrations of various fatty

acids, we performed MS-based targeted analysis of serum. In agreement with the global lipidomics results, circulating levels of THA were markedly elevated in Acox1-LKO mice, while EPA and DHA were decreased. Various long and medium chain fatty acids were also decreased (Supplementary Fig. 3d), consistent with the compensatory increase in expression of mitochondrial fatty oxidation genes. Measurement of hepatic fatty acid composition also revealed a marked increase in THA in Acox1-LKO mice (Supplementary Fig. 3e), supporting the notion that the elevated circulating levels of this fatty acid reflects its increased hepatic accumulation and secretion in Acox1-LKO mice. Together, these results suggest that inhibition of peroxisomal β-oxidation and increased expression of fatty acid elongation and desaturation genes leads to the accumulation of ω-3 VLCFA, especially THA, in the liver (Supplementary Fig. 3f). Liver-derived acylcarnitines have been reported to serve as an alternative fuel source for BAT-mediated thermogenesis[23]. However, our lipidomic analysis showed a few species of acylcarnitine were increased while others were decreased in Acox1-LKO mouse serum. (Supplementary Fig. 3g). Given that the altered acylcarnitine species were present in low abundance, it is unlikely that the iWAT browning in Acox1-LKO mice is mediated by circulating acylcarnitines.

## Very long chain ω-3 fatty acids promote thermogenic gene expression and improve insulin sensitivity in adipocytes

We next investigated the hypothesis that the beneficial effects of hepatic Acox1 inactivation, including adipocyte browning, increased mitochondrial respiration and improved insulin sensitivity, are mediated by increased circulating ω-3 VLCFAs. The role of conventional ω-3 fatty acids in adipocyte browning is somewhat controversial[24], although multiple groups have reported that higher concentrations (100-200 μM) of EPA and α-linolenic acid (ALA; C18:3, n-3), but not DHA, promote thermogenic gene expression[25,26]. However, the physiological role of ω-3 VLCFAs, such as THA and HHA, has not been studied. To investigate the role of these fatty acids in adipocyte browning and compare their effects to those of EPA, we induced differentiation in iWAT SVF cells and treated the cells with 50 μM THA, HHA, EPA, or vehicle (Fig. 4a). Treatment of iWAT cells with THA, the major species upregulated in Acox1-LKO mice, did not affect adipogenesis (Fig. 4b). However, lipid droplets in the fatty acid-treated cells were noticeably smaller (Fig. 4c), indicating potential activation of browning. Consistent with this possibility, THA and HHA significantly increased the expression of multiple brown and beige fat genes (Fig. 4d). In contrast, 50 μM EPA did not uniformly increase the expression of thermogenic genes, suggesting that VLCFA ω-3 fatty acids are more potent inducers of adipocyte browning. Western blot and immunofluorescence analyses showed that THA and HHA also markedly increased protein expression of the mitochondrial proteins UCP1 and COX4 (Fig. 4e, f). In preadipocytes derived from human WAT, THA or HHA treatment also increased the expression of browning markers (Supplementary Fig. 4a, b), without affecting adipogenesis (Supplementary Fig. 4c).

Given the increased expression of mitochondrial proteins, we determined whether THA promotes mitochondrial biogenesis. Quantitative real-time PCR analysis revealed that THA increases expression of mitochondrial biogenesis factors Tfam and Nrf1 and mitochondrial DNA (mtDNA)-encoded oxidative phosphorylation (OXPHOS) pathway components cytochrome c oxidase I and II (MtCo1 and MtCo2), and mitochondrially encoded NADH:ubiquinone oxidoreductase core subunit 6 (MtNd6) (Fig. 4g). Subsequent experiments were done primarily with THA, since it was the predominant species upregulated in serum from Acox1-LKO mice (Fig. 3g). Consistent with activation of mitochondrial biogenesis, mtDNA copy number normalized to nuclear DNA was significantly increased with THA treatment (Fig. 4h). Accordingly, assessment of mitochondrial bioenergetics using a Seahorse XF24 Extracellular Flux Analyzer revealed that THA treatment

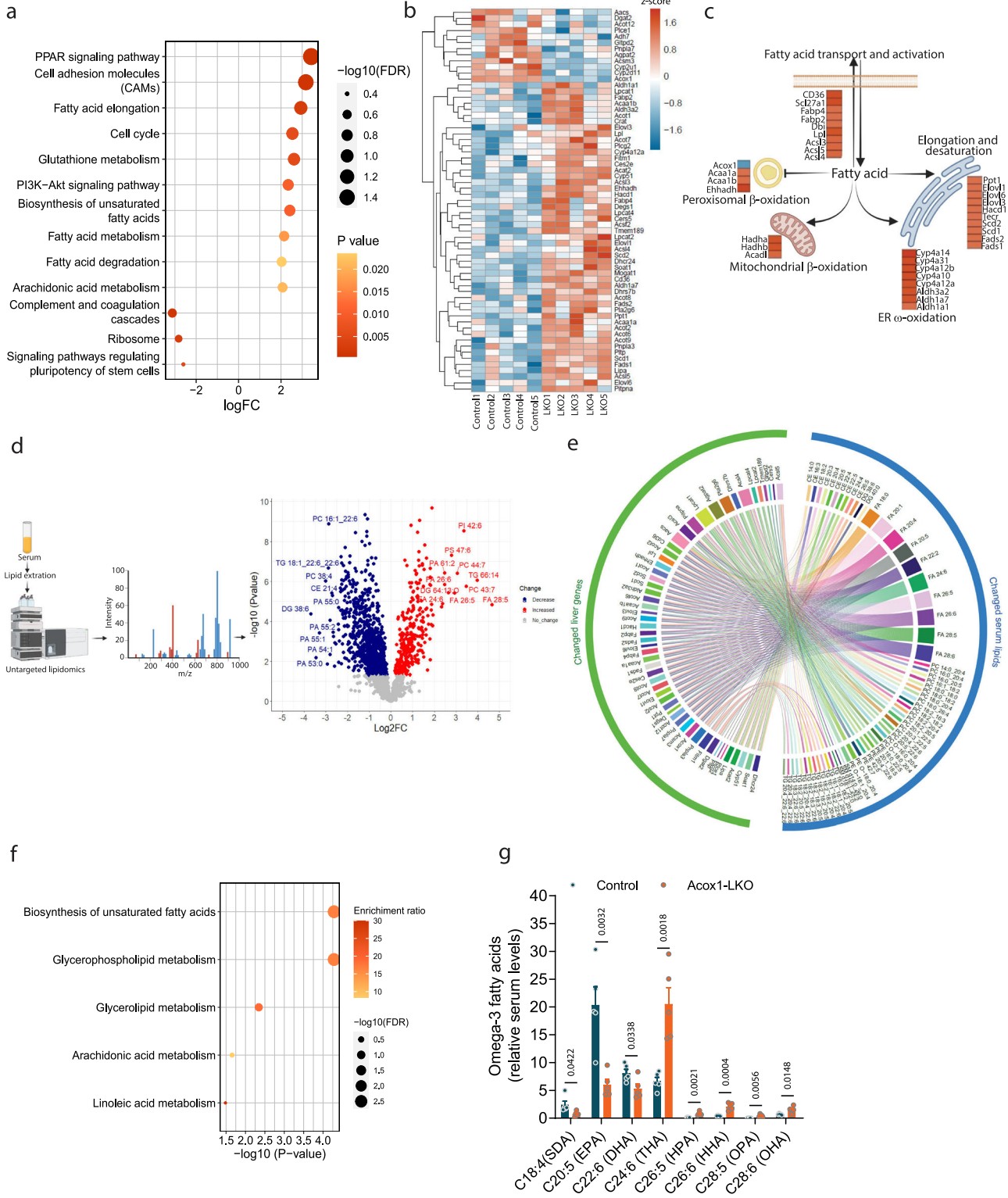

**Fig. 3 | Upregulation of liver-derived ω−3 very long chain fatty acids in the circulation of *Acox1*-LKO mice. a** The top changed KEGG pathways in RNA-seq analysis in livers of control and *Acox1*-LKO mice. **b** Heatmap of all changed lipid metabolic genes in RNA-seq analysis (*n* = 5). **c** Mapping of changed lipid metabolic genes to fatty acid metabolic processes: fatty acid transport and activation, peroxisomal β-oxidation, mitochondrial β-oxidation, ER ω-oxidation and elongation and desaturation. Created with BioRender.com. **d** Untargeted lipidomic analysis of serum from control and *Acox1*-LKO mice (*n* = 5). Schematic created with

BioRender.com. **e** Circos plot depicting relationship between liver transcriptomic changes and serum lipidomic changes in *Acox1*-LKO mice. **f** Pathway enrichment analysis of changed lipids using MetaboAnalyst. **g** Significantly altered ω−3 fatty acids identified by serum lipidomics (*n* = 5). Data in (**b**, **d**, **g**) are from biologically independent samples. Data with error bars are reported as the mean ± SEM. *P* values were determined by one-tailed Fisher's exact test in (**a**) and (**f**) or two-sided unpaired Student's *t* test in (**d**) and (**g**). Source data are provided as a Source Data file.

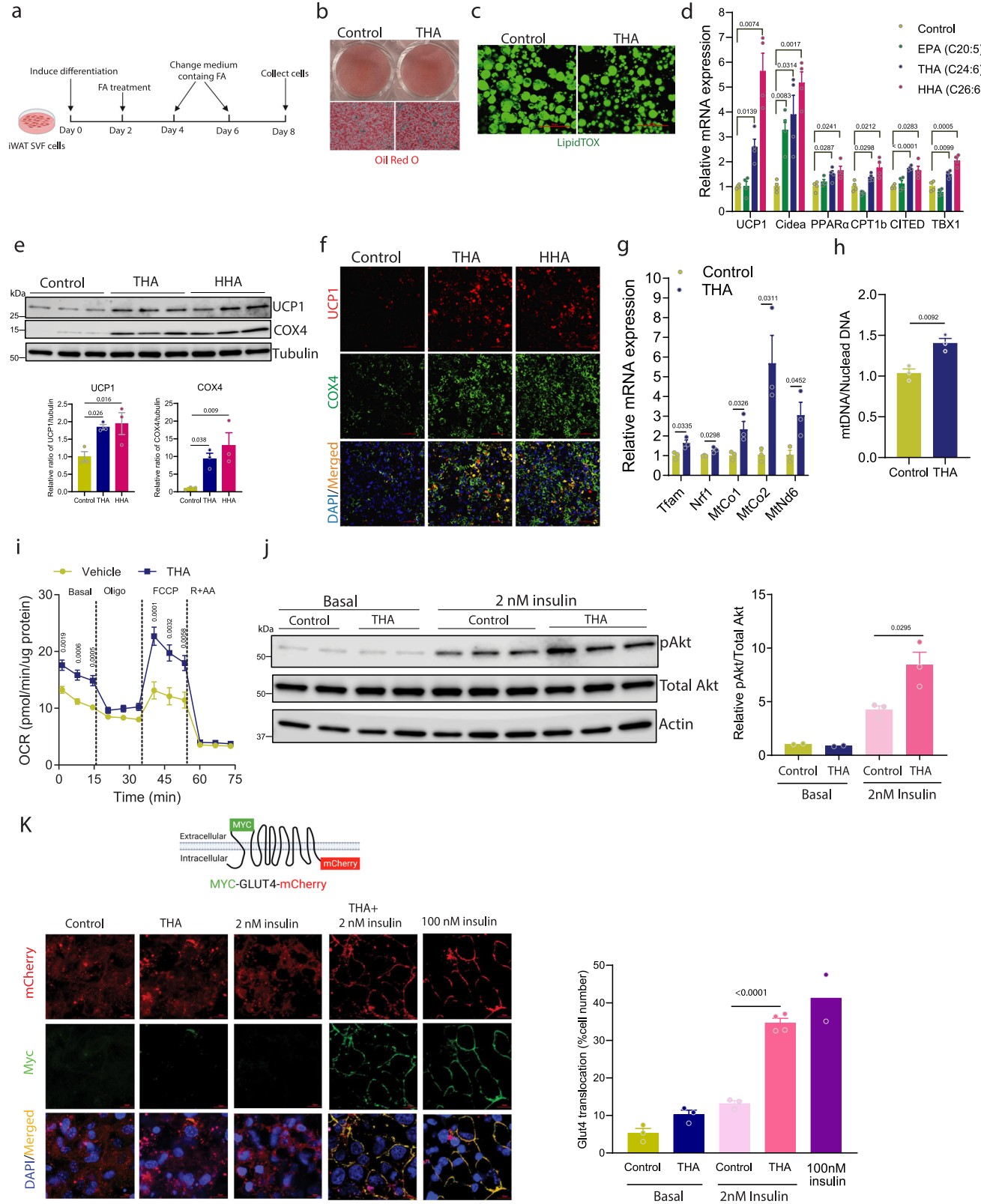

increases uncoupled respiration in mouse and human white adipocytes (Fig. 4i and Supplementary 4d).

Since *Acox1*-LKO mice display improved insulin sensitivity, we also determined the effect of THA on insulin-stimulated Akt phosphorylation and translocation of the Glut4 glucose transporter in adipocytes. Although THA alone had no effect on Akt phosphorylation in serum-starved cells, it increased the phosphorylation in adipocytes treated with 2 nM insulin (Fig. 4j). THA or 2 nM insulin alone did not significantly increase plasma membrane translocation of Glut4, but THA potentiated the effect of the low dose insulin on translocation of the glucose transporter, resembling the effect of 100 nM insulin (Fig. 4k). Together, these results support the possibility that ω-3 VLCFAs, especially THA, mediate the improved insulin sensitivity and adipocyte browning observed in *Acox1*-LKO mice.

**Fig. 4 | ω-3 Very long-chain fatty acids induce iWAT adipocyte browning and Glut4 translocation. a** Strategy for treatment of iWAT SVF cells with fatty acids. Created with BioRender.com. **b** Oil Red O staining in control and THA-treated iWAT SVF cells (scale bar, 75 μm). **c** LipidTox staining of lipid droplets in control and THA-treated iWAT SVF cells (scale bar, 50 μm). **d** Expression of browning marker in adipocytes treated with vehicle, EPA, THA or HHA ($n = 4$). **e** UCP1 and COX4 protein levels in adipocytes treated with vehicle, THA or HHA ($n = 3$). **f** Immunofluorescence analysis of UCP1 and COX4 in differentiated iWAT SVF treated with THA, HHA or vehicle (scale bar, 50 μm). **g** Expression of mitochondrial gene in adipocytes treated with vehicle or THA ($n = 3$). **h** Mitochondrial DNA copy number in adipocytes treated with vehicle or THA ($n = 3$). **i** OCR of adipocytes treated with vehicle or THA ($n = 7$). **j** Basal and insulin activated Akt phosphorylation in adipocytes treated with vehicle or THA ($n = 2$ for basal, 3 for insulin treatment). **k** Glut4 translocation in adipocytes treated with vehicle or THA in the presence of 0, 2 nM or 100 nM insulin (scale bar, 25 μm). Quantification of Glut4 translocation is based on $n = 3$/group (basal), 3 (control+ 2 nM insulin), 4 (THA + 2 nM insulin), and 2 (100 nM insulin). Data in (**d**, **e**) and (**g**–**i**) are from biologically independent samples. Data with error bars are reported as the mean ± SEM. $P$ values were determined by two-sided unpaired Student's $t$ test in (**g**, **h**, **j**) (right panel), two-way ANOVA followed by Fisher's LSD test in (**d**, **k**) (right panel) or one-way ANOVA followed by Fisher's LSD test in (**e**) (bottom panel). Source data are provided as a Source Data file.

## THA induces browning through activation of GPR120

We next sought to understand the molecular mechanism through which ω-3 VLCFAs promote adipocyte browning. The ligand-activated transcription factor PPARγ is the master regulator of adipocyte gene expression, including the expression of thermogenic genes. PPARγ has been reported to be activated by a variety of lipids, including polyunsaturated fatty acids[27]. We thus determined whether ω-3 VLCFAs are endogenous agonists of PPARγ using a PPARγ-dependent luciferase reporter assay. In contrast to low dose rosiglitazone, which expectedly induced PPARγ activation, 50 μM of THA or HHA, a concentration that promotes adipocyte browning, did not significantly activate PPARγ (Supplementary Fig. 5a).

We next investigated whether these fatty acids might induce adipocyte browning through activation of a family of G-protein coupled receptors (GPCRs) called free fatty acid receptors (FFARs). At least five FFARs and other fatty acid-binding GPCRs have been described: GPR40 (FFAR1), GPR41 (FFAR2), GPR43 (FFAR3), GPR120 (FFAR4), and GPR84. To understand the importance of these receptors in adipose tissue browning in mice, we first determined the effect of cold exposure on their gene expression in iWAT. Only GPR43 (*Ffar3*) and GPR120 (*Ffar4*) expression significantly increased after cold exposure, with GPR120 (*Ffar4*) showing a greater fold change (Fig. 5a). GPR40 and GPR120 are activated by medium and long chain fatty acids, such as palmitate, oleate, and ω-3 fatty acids EPA and DHA[28,29]. GPR41 and GPR43 are activated by short-chain fatty acids, such as butyrate, propionate, and acetate[30], while GPR84 is activated by medium-chain fatty acids[31]. It is unknown if these receptors can be activated by ω-3 VLCFAs, such as THA. To this end, we used the PRESTO-Tango assay[32], which is based on GPCR activation-mediated recruitment of β-arrestin inducing expression of a tTA-dependent luciferase reporter (Fig. 5b). Using this assay, we discovered that THA selectively activates GPR120 among the five different fatty acid binding GPCRs (Fig. 5c). To further confirm that THA is a GPR120 ligand, we used a BRET-based GPR120 binding assay (Supplementary Fig. 5b). Our results show that THA significantly increases GPR120-dependent BRET signal, with potency similar to EPA (Supplementary Fig. 5c). GPR120 is a $G_q/G_{11}$-coupled receptor that promotes intracellular calcium mobilization. To determine if THA promotes calcium release, we ectopically expressed GPR120 in 293 T cells which were then stained with a Fluo-4 calcium indicator. THA treatment strikingly promoted calcium accumulation and was more potent than EPA (Fig. 5d, e).

Next, we determined whether the ω-3 VLCFA-induced browning was GPR120 dependent. Inhibition of GPR120 using a selective antagonist, AH7614, significantly decreased THA-induced increase in thermogenic genes in iWAT-derived adipocytes (Fig. 5f). Similarly, knockdown of GPR120 using lentivirus-delivered shRNA decreased THA-induced increase in *Ucp1* gene expression (Fig. 5g) and protein levels (Fig. 5h). Accordingly, the THA-induced increase in uncoupled respiration was significantly decreased by GPR120 (*Ffar4*) knockdown (Fig. 5i). To further assess a link between liver ACOX1 and adipose tissue GPR120, we performed gene expression analysis of GPR120 (*Ffar4*) in *Acox1*-LKO and control mice fed chow diet or a HFD.

Although high-fat feeding nearly significantly decreased GPR120 (*Ffar4*) gene expression in iWAT of control mice, hepatic *Acox1* inactivation increased the expression of the lipid sensor (Supplementary Fig. 5d). Together, these data suggest that circulating ω-3 VLCFAs, which accumulate due to liver *Acox1* knockout, promote adipocyte browning through activation of GPR120.

## Pharmacological inhibition of ACOX1 promotes adipose tissue browning and protects mice against diet-induced obesity

We next addressed the translational relevance of inhibiting ACOX1 to improve metabolic health. To this end, we determined whether pharmacological inhibition of ACOX1 mimics the effects of genetic inactivation of the enzyme. 10,12-Tricosadiynoic acid (TDYA) is a highly specific inhibitor of ACOX1 that acts as a suicide substrate[33]. Structurally, TDYA consists of a 23 carbon chain, thus resembling a VLCFA, but has two triple bonds at positions Δ10 and Δ12 (Supplementary Fig. 6a) which prevent its oxidization. Using $D_3$-C22:0, a stable isotope-labeled VLCFA, we found that TDYA treatment in AML12 cells leads to significantly decreased levels of the chain-shortened products $D_3$-C18:0 and $D_3$-C16:0 in AML12 cell (Supplementary Fig. 6b), consistent with the notion that TDYA inhibits VLCFA β-oxidation. Consistent with the inhibition of ACOX1 activity, TDYA treatment significantly increased the liver levels of VLCFAs, including THA (Supplementary Fig. 6c). However, DHA was not decreased with TDYA treatment as it was with genetic inactivation of *Acox1*, suggesting that the pharmacological inhibitor has a more complex effect on VLFA levels.

Previous studies indicate that TDYA supplementation decreases high-fat diet-induced obesity and hepatic steatosis in rats[33]. However, these results were confounded by the fact that the inhibitor was administered daily by oral gavage for 8 weeks, which is stressful for animals. Moreover, the underlying molecular mechanism remains unclear. Thus, we fed wild-type C57BL/6J mice HFD containing 50 mg/kg TDYA or control HFD without TDYA. Like *Acox1*-LKO mice, TDYA-treated animals had significantly decreased body weight (Supplementary Fig. 6d). Histologic analysis indicated that adipocytes in iWAT were significantly smaller (Supplementary Fig. 6e, f). The decreased adiposity was not due to altered food intake (Supplementary Fig. 6g) or physical activity (Supplementary Fig. 6h). Instead, indirect calorimetry indicated that energy expenditure trends higher in TDYA-treated animals, especially during the dark cycle (Supplementary Fig. 6i). Since the indirect calorimetry was performed in mice that already had a body weight difference between the genotypes, we also analyzed the results using analysis of covariance (ANCOVA), with body weight as a covariate. The regression-based analysis confirmed that ACOX1 inhibition promotes a significant mass-independent increase in energy expenditure (Supplementary Fig. 6j). This led us to check whether adipose tissue browning is increased in the setting of high-fat feeding, as in *Acox1*-LKO mice. Gene expression analysis showed that brown adipocyte genes, such as *Ucp1*, were significantly increased in iWAT of TDYA-treated mice (Supplementary Fig. 6k). Western blot analysis showed that TDYA treatment results in increased levels of UCP1 and Mitochondrial Complex II (Supplementary Fig. 6l).

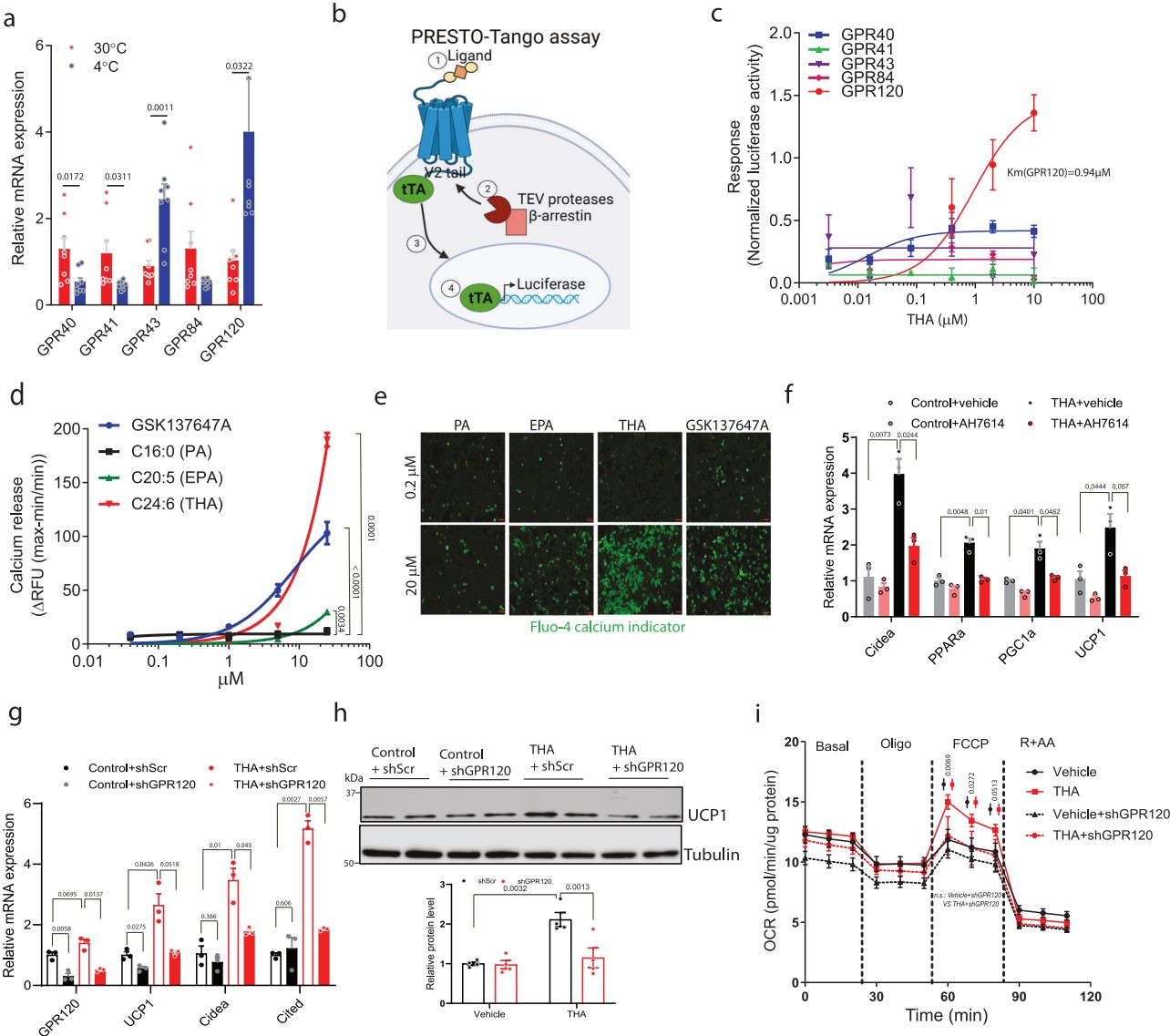

**Fig. 5 | THA selectively activates GPR120 to promote adipocyte browning.**
**a** Gene expression of five fatty acid receptors in iWAT of WT mice housed under 30 °C or 4 °C ($n = 8$). **b** Schematic representation of Tango (β-arrestin recruitment) assay: (1) Activation of GPCR by agonist. (2) Recruitment of β-arrestin and TEV protease to the C terminus. (3) Cleavage of TEV site leading to the release of the transcription factor tTA. (4) Released tTA induces luciferase expression. Created with BioRender.com. **c** Tango assay of five fatty acid receptors stimulated by THA ($n = 4$). **d** Calcium release assay in 293T cells expressing mouse GPR120 after treatment with GSK137647A, palmitic acid (PA), EPA or THA ($n = 4$).
**e** Representative images of cell stained with Fluo-4 calcium indicator after

treatment with GSK137647A, PA, EPA or THA at 0.2 μM or 20 μM (scale bar, 50 μm).
**f** Gene expression in iWAT adipocytes treated with PA or AH, in the presence of GPR120 inhibitor AH7614 or vehicle ($n = 3$). **g–i** Gene expression (**g**), UCP1 protein levels (**h**) and OCR (**i**) of Lenti-Scr or Lenti-shGPR120-infected iWAT adipocytes treated with vehicle or THA; ($n = 3$ in (**g**), $n = 4$ in (**h**) and $n = 11$ in (**i**)). Data in (**a, c, d**) and (**f–i**) are from biologically independent samples. Data with error bars are reported as the mean ± SEM. $P$ values were determined by two-sided unpaired Student's $t$ test in **a** or two-way ANOVA followed by Fisher's LSD test in (**d**), and (**f–i**). n.s., not significant. Source data are provided as a Source Data file.

Next, we investigated the effect of inhibiting ACOX1 activity on glucose homeostasis and insulin sensitivity. TDYA treatment significantly improved glucose tolerance in HFD-fed mice (Supplementary Fig. 6m). Consistent with the improved insulin sensitivity, serum insulin levels were nearly significantly decreased (Supplementary Fig. 6n), while adiponectin was significantly increased (Supplementary Fig. 6o). Accordingly, TYDA increased the insulin-stimulated Akt phosphorylation in iWAT and gWAT (Supplementary Fig. 6p). Together, these results suggest pharmacological inhibition of ACOX1 phenocopies liver-specific knockout of the peroxisomal β-oxidation enzyme. However, the effects of genetic and pharmacological inhibition of ACOX1 were not identical. For example, the body weight difference in TDYA-treated mice is smaller and emerges later as

compared to Acox1-LKO animals. This is presumably because TDYA-mediated inhibition of ACOX1 activity is partial and not liver-specific as opposed to genetic ablation of *Acox1*. Moreover, the pharmacological inhibitor may need more time to accumulate in tissues and target the enzyme. Nevertheless, the two models resemble in several aspects, including decreased diet-induced obesity, increased thermogenic gene expression and improved glucose homeostasis.

**THA promotes adipose tissue browning in mice**
Finally, we investigated whether THA, the predominant circulating fatty acid that increases as a result of inhibiting ACOX1, is sufficient to improve metabolic health in obese mice. To this end, we fed WT C57BL/6J mice an HFD for 9–11 weeks to induce obesity and insulin

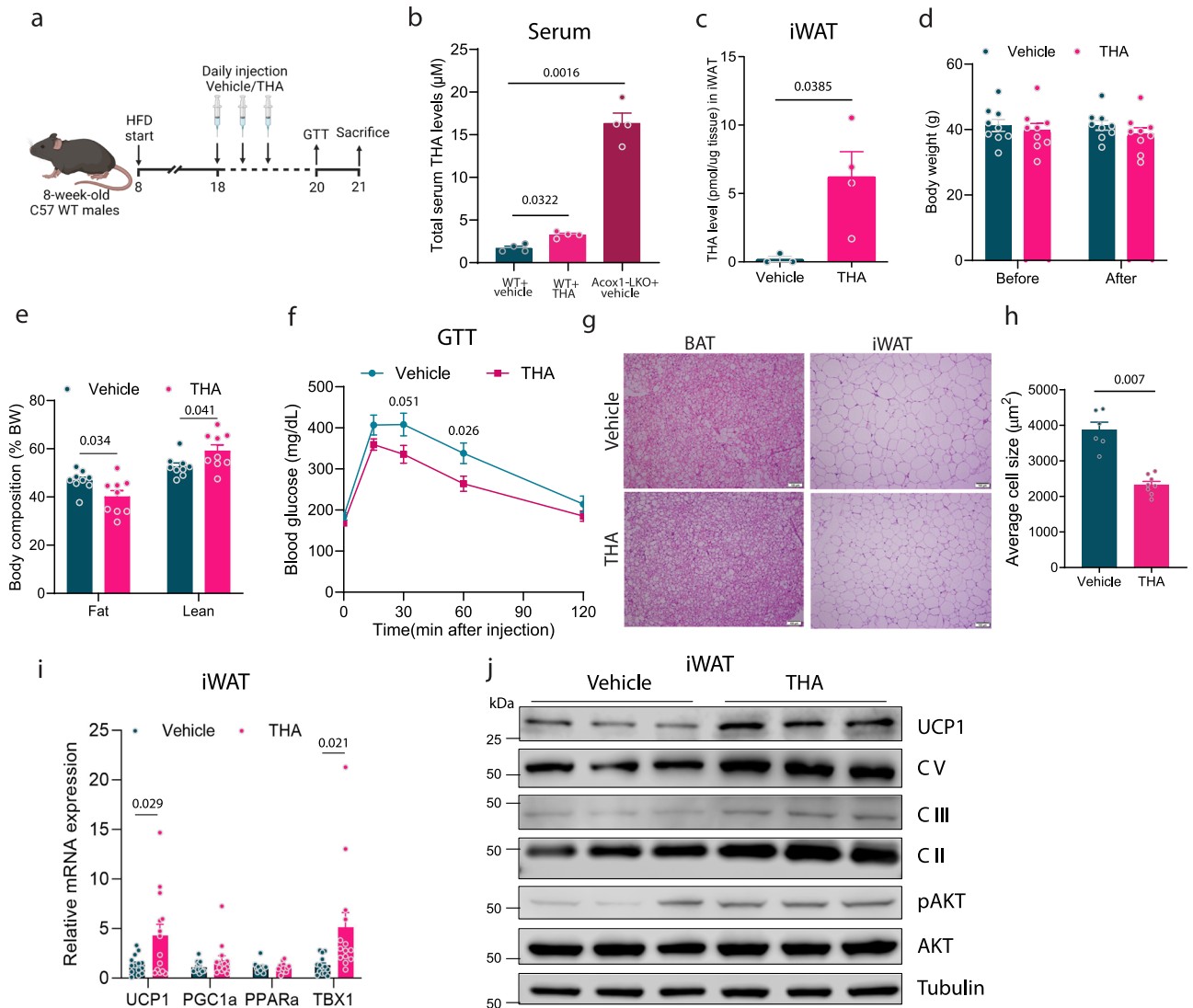

**Fig. 6 | THA promotes adipose tissue browning in mice. a** Schematic of THA treatment in mice. Created with BioRender.com. **b** Serum THA levels in vehicle-treated WT mice, THA-treated WT mice, or vehicle-treated *Acox1*-LKO mice ($n = 4$). **c** Free THA level in iWAT of mice injected with vehicle or THA for 15 min, $n = 3$ (vehicle), $n = 4$ (THA). **d** Body weight of mice before and after vehicle or THA injections ($n = 9$). **e** MRI analysis of mice after vehicle or THA treatment ($n = 9$). **f** Glucose tolerance test of control and THA-treated mice ($n = 14$). **g** H&E staining of BAT and iWAT from control or THA-treated mice (scale bar, 100 μm).

**h** Quantification of iWAT cell size, $n = 6$ (vehicle), $n = 8$ (THA). **i** Brown gene expression in iWAT of control or THA treated mice ($n = 14$). **j** Western blot analysis of UCP1, OXPHOS proteins and AKT phosphorylation in iWAT tissue ($n = 3$). Data with error bars are reported as the mean ± SEM. *P* values were determined by one-way ANOVA followed by Dunnett's multiple comparisons test in (**b**) or two-sided unpaired Student's *t* test in (**c**–**e**, **h**), and (**i**), or two-way ANOVA followed by Fisher's LSD test in (**f**). Source data are provided as a Source Data file.

resistance and then treated the animals with THA or vehicle by daily intraperitoneal (i.p.) injection for 2 weeks while continuing the obesogenic diet (Fig. 6a). Measurement of serum concentration of THA by mass spectrometry indicated that the circulating levels of the i.p. administered THA were significantly increased compared to the vehicle-treated mice, but markedly lower than the levels of the endogenous lipid in *Acox1*-LKO mice (Fig. 6b), presumably due to metabolism of the exogenous THA by hepatic peroxisomal β-oxidation prior to reaching the systemic circulation in WT mice. We also determined the adipose tissue levels of THA 15 min after i.p. treatment and observed that the fatty acid was dramatically increased above the baseline levels in iWAT (Fig. 6c), confirming that the exogenously supplied fatty acid reaches the target tissue rapidly after i.p. treatment. During the short-term (2-week) treatment in HFD-fed animals, THA did not affect the body weight (Fig. 6d). However, fat mass normalized to body weight was significantly decreased while lean mass was increased (Fig. 6e) and glucose tolerance was improved (Fig. 6f). Histologic

analysis revealed that adipocytes in iWAT were smaller, while there was no difference in BAT morphology (Fig. 6g). Quantification of cell size showed that iWAT adipocytes were significantly smaller in THA-treated mice (Fig. 6h). Gene expression analysis indicated that THA treatment increases thermogenic genes, including *Ucp1*, in iWAT (Fig. 6i). Western blot analysis confirmed the increased expression of UCP1 and mitochondrial respiratory complex proteins in iWAT and demonstrated that THA promotes Akt phosphorylation (Fig. 6j). Together, these results indicate that THA, one of the circulating fatty acids that increases in *Acox1*-LKO mice, is sufficient to promote adipose tissue browning and mediate the improved glucose homeostasis associated with liver-specific inactivation of peroxisomal β-oxidation.

## Discussion

Peroxisomes are involved in β-oxidation of VLCFAs, a substrate that cannot be catabolized in mitochondria. Whether liver peroxisomal β-oxidation influences systemic metabolism to affect diet-induced

obesity has not been investigated. Our results identify hepatic peroxisomal β-oxidation as an important regulator of whole-body metabolic homeostasis. Liver-specific knockout of the peroxisomal β-oxidation enzyme ACOX1 promotes resistance to diet-induced obesity, decreases infiltration of pro-inflammatory macrophages into adipose tissue in male mice, and improves systemic glucose homeostasis in both male and female mice. To understand potential mechanisms through which liver-specific knockout of *Acox1* promotes adipose tissue browning and metabolic health, we used a combination of liver transcriptomics and untargeted serum lipidomics. Our results indicate that inhibition of hepatic peroxisomal β-oxidation results in increased circulating levels of polyunsaturated VLCFAs, such as THA (C24:6) and HHA (C26:6), likely due to a combination of impaired peroxisomal β-oxidation and increased expression of genes involved in fatty acid elongation and desaturation. We further show that these fatty acids promote adipocyte browning and metabolic health through their role as endogenous ligands of GPR120. Together, these studies reveal a previously unknown liver-adipose signaling axis that improves metabolic homeostasis.

Increasing evidence shows that lipids are not just the preferred fuel source for adipose tissue thermogenesis, but are also involved in mediating inter-organ communication, crosstalk between organelles, and other signaling roles that affect adipose tissue thermogenesis[34,35]. Our results point toward a signaling role related to adipocyte browning for the ω-3 VLCFAs that accumulate in the serum of *Acox1*-LKO mice. Previous studies indicate that the conventional ω-3 fatty EPA, but not DHA, promotes thermogenic gene expression in adipocytes, though the effects of EPA are somewhat inconsistent and higher concentrations are required[25,26]. Our results show that THA and HHA robustly promote thermogenic gene expression in adipocytes derived from iWAT, in contrast to the modest effects observed using EPA. Moreover, the ω-3 VLCFAs promote mitochondrial biogenesis. Together, these results suggest that these previously uncharacterized fatty acids are potent inducers of adipocyte browning.

Adipose tissue browning is mediated by PPARγ activation through stabilization of the thermogenic transcriptional co-regulator PRDM16[36]. Although PPARγ is activated by a broad range of fatty acids, including polyunsaturated fatty acids[27], THA or HHA did not promote PPARγ transcriptional activity. Instead, our results indicate that THA, the predominant circulating fatty acid increased in *Acox1*-LKO mice, activates GPCR signaling. Of the five known free fatty acid binding GPCRs, THA selectively activated GPR120 (FFAR4), which is known to bind ω-3 fatty acids, including EPA and DHA[37]. Our results suggest that THA is a high-affinity ligand of GPR120, whose circulating levels are increased while those of EPA and DHA are decreased in *Acox1*-LKO mice. Together, this work identifies THA as a novel endogenous ligand of GPR120.

GPR120 is related to obesity in humans. The adipose tissue gene expression of GPR120 is elevated in humans with obesity as compared to lean individuals. Moreover, a loss of function GPR120 mutation (R270H) is associated with increased risk for obesity in European populations[38]. Several aspects of the liver-specific *Acox1* knockout support the concept that hepatic peroxisomal β-oxidation regulates endogenous activation of GRP120 to impact adipose tissue browning and metabolic health. The circulating ω-3 VLCFAs that accumulate in *Acox1*-LKO mice promote adipocyte browning, mitochondrial biogenesis and Glut4 translocation, similar to the effects of known activators of GPR120[37]. While GPR120 (*Ffar4*) knockout mice fed a high-fat diet have an increased propensity to develop obesity and glucose intolerance[38], *Acox1*-LKO mice exhibit adipose tissue browning, decreased diet-induced obesity, and improved systemic glucose homeostasis, all consistent with increased GPR120 activation. Obesity is associated with low-grade chronic inflammation that contributes to systemic insulin resistance[13]. High-fat diet-fed *Acox1*-LKO mice resist infiltration of pro-inflammatory immune cells in visceral adipose tissue, similar to the effect of treating obese mice with a small-molecule GPR120-selective agonist[39].

Although prenatal global knockout of *Acox1* in the 129/Sv mouse strain leads to severe abnormalities, including growth retardation, hepatic steatosis, and hepatocellular carcinoma[10,40], our work shows that mice with liver-specific knockout of *Acox1* on the C57BL/6 J genetic background are healthy with normal liver function. The phenotypic difference between our model and the global knockout mice might reflect differences in genetic backgrounds, since 129/Sv mice are known for having increased susceptibility for developing cancer[41]. It also possible that extrahepatic deletion of *Acox1* in the global knockout mouse model contributes to liver disease. However, our results indicate that systemic inhibition of ACOX1 activity in mice using a pharmacological inhibitor does not lead to abnormalities and instead improves metabolic health. Thus, ACOX1 is a potentially druggable target.

In summary, this work identifies hepatic peroxisomal β-oxidation as an important regulator of metabolic homeostasis that limits circulating levels of endogenously produced ω-3 VLCFAs that influence the energy expenditure and metabolic functions of adipose tissue through activation of GPR120 signaling. Dietary supplementation of these beneficial lipids or inhibition of ACOX1 enzyme activity has the potential to be an effective strategy for treating obesity-associated metabolic disorders.

## Methods
### Mouse models and animal experiments
All animal protocols were approved by the Washington University Institutional Animal Care and Use Committee. *Acox1* liver-specific knockout mice on the C57BL/6 J genetic background were generated as previously described[42]. C57BL/6J wild-type mice were obtained from the Jackson Laboratory (Strain #000664). All mice were maintained under constant temperature, circulating air and 45-65% humidity with 12 h:12 h light/dark cycle. Mice had free access to chow diet (Purina 5053) and water. For the HFD treatment, *Acox1* floxed mice without Cre were used as the WT control. Eight weeks old control and *Acox1*-LKO mice were fed a 60% HFD (D12492, Research Diets). ACOX1-inhibitor diet was generated by adding 10,12-tricosadiynoic acid (Sigma #91445) to the D12492 HFD at a concentration of 50 mg/kg. For cold exposure, mice were housed at 4 °C for 48 h. For C24:6 fatty acid treatment, WT mice were fed a 60% HFD for 9 weeks to induce obesity. They were then injected daily with 25 mg/kg body weight of THA or vehicle for two weeks. Metabolic phenotyping experiments were primarily conducted using male animals. In selected experiments where female mice were used, the sex is specified in the figure legend. All phenotypic data were disaggregated by sex. Mice were euthanized by i.p. injection of a cocktail containing 500 mg/kg ketamine and 50 mg/kg xylazine.

Glucose tolerance tests were performed on mice that had been fasted for 6 h by intraperitoneal injection of 1 g/kg dextrose. Blood glucose levels were measured using a blood glucose test meter from Glucocard (Arkray USA. Inc.) at 0, 15, 30, 60, and 120 min after injection. Insulin tolerance tests were performed on mice that had been fasted for 4 h by injecting them with 0.75 U/kg insulin (Humulin R, Lilly). Blood glucose was measured at 0, 15, 30, 60, and 120 min after injection.

### Characterization of immune cells in adipose tissue by flow cytometry
Mice were perfused through right ventricle with 30 mL of ice-cold PBS. Gonadal adipose tissue was collected and minced to small pieces. Then the tissues were digested for 30 min at 37 °C in phenol red-free DMEM + 0.5% BSA + collagenase D (1 mg/mL; Roche #11088882001). Upon digestion, cells were sieved through 100 μm cell strainers and spun down at 500 × *g*, 10 min, 4 °C without break. Supernatants were carefully aspirated and the pellet containing stromal vascular fraction

(SVF) was incubated with red blood cell lysis buffer (Gibco #A1049201) for 3 min. 10 ml of cold DMEM was added and cells were spun at 500 × g, 10 min, 4 °C. Cells were then resuspended in PBS and counted to obtain total SVF counts in both fat pads from each mouse.

(2–3) × 10⁵ cells were incubated with Live-or-Dye 665/685 viability dye (1000x diluted according to manufacturer's instruction; Biotium #32013) in PBS for 15 min at 4 °C, in 96-well U-bottom plate. Cells were washed and blocked with TruStain FcX PLUS (2.5 µg/mL; Biolegend #156604) prepared in FACS buffer (DPBS, 2% FBS, 2 mM EDTA). Cells were then stained for 30 min at 4 °C with the murine cocktail of antibodies CD45 BV480, CD3 BV650, CD4 PE/Dazzle 594, CD8a PE, TCR g/d APC, CD19 BV711, F4/80 PE/Cyanine7, CD11b BV605, CD9 BV786, Ly-6C BV570, CD11c PE/Cyanine5, CD206 Alexa Fluor 700, I-A/I-E Spark Blue 550, NK-1.1 BV750, Ly-6G FITC, FceR1a BV510, XCR1 BV421, siglec-F APC-Cy7, CD44 PerCP/Cyanine5.5. Cells were washed 3 x with FACS buffer, resuspended in ice-cold PBS and acquired on Cytek Northern Lights 3-laser spectral cytometer.

The unmixing algorithm was performed using SpectroFlo (Cytek). FCS files were then analyzed using OMIQ analysis platform. Doublets, debris, and dead cells were excluded, and live CD45+ were subsampled to 48774 cells/ mouse and analyzed. Uniform manifold approximation and projection (UMAP)[43] was used to visualize single-cell data. Flow-SOM algorithm was used to generate filters, which were then manually inspected and adjusted in some cases. A total number of cells in identified populations was calculated from total SVF cell counts and CD45 percentages in each sample (mouse). Expression of CD11c and CD206 surface markers was used to evaluate ATM remodeling.

## Cell lines and treatments
AML12 (alpha mouse liver 12) cells were obtained from ATCC and cultured in DMEM/F12 containing 10% FBS, 1x Insulin-Transferrin-Selenium (ITS -G, Gibco, #41400045) and 40 ng/ml dexamethasone. Human embryonic kidney 293 T (HEK293T) cells were maintained in DMEM-high glucose (Sigma #D6429) supplemented with 10% FBS. HTLA cells (HEK293 cells stably expressing the β-arrestin2-TEV fusion gene and tTA-dependent luciferase reporter) used for GPCRs β-arrestin recruitment Tango assay have been reported previously[44]. HTLA cells were grown in DMEM supplemented with 10% FBS, 1% penicillin/streptomycin, 2 µg/ml puromycin, and 100 µg/ml hygromycin. iWAT SVF cells were isolated from 6-week-old C57BL/6 J mice, immortalized, and differentiated into adipocytes as previously reported[45]. Briefly, confluent SVF cells were treated with DMEM/F12 supplemented with 0.5 µM isobutylmethylxanthine (Sigma #I5879), 5 µM dexamethasone (Sigma #D-1756), 125 µM indomethacin (Sigma #I7378), 1 µM rosiglitazone (Sigma #71740), 1 nM T3 (Sigma #T2877), and 0.02 µM insulin (Sigma #I-5523) for 2 days. Then, the cells were switched to medium supplemented with 1 nM T3 and 0.02 µM insulin (maintenance medium). The immortalized human white adipose tissue preadipocyte (hWAT) cell line was kindly provided by Dr. Yu-Hua Tseng (Joslin Diabetes Center) and have been previously described[46]. A modified protocol was used to differentiate hWAT preadipocytes into adipocytes. The cells were cultured to confluence for 6 days, then treated with culture medium containing 33 µM biotin (Sigma #B4501), 0.5 µM human insulin, 17 µM pantothenate (Sigma #C8731), 0.1 µM dexamethasone, 2 nM T3, 500 µM isobutylmethylxanthine, and 30 µM indomethacin for 18 days. To treat differentiated mouse iWAT SVF cells with mouse serum, the maintenance medium was changed to DMEM/F12 supplemented with 10% serum from control or Acox1-LKO mice after 6 days of differentiation and then maintained for 2 days.

Palmitic acid, eicosapentaenoic acid (Cayman #90110), 6(Z), 9(Z), 12(Z), 15(Z), 18(Z), 21(Z)-tetracosahexaenoic acid (Larodan #10-2406), and 8(Z), 11(Z), 14(Z), 17(Z), 20(Z), 23(Z)-hexacosahexaenoic acid (Larodan #10-2606) were used to treat cells at a concentration of 50 µM throughout the differentiation process. AH7614 (Cayman Chemical) was used to treat differentiated iWAT SVF cells at a concentration of 100 µM.

## Plasmids
The GPR120 shRNA was obtained from Sigma. pLenti-myc-GLUT4-mCherry was from Addgene. To construct the pLJM1-GPR120 plasmid, GPR120 was amplified by PCR using WT mouse iWAT cDNA, and then the amplicon was cloned into pLJM1-EGFP (Addgene #19319) using AgeI and EcoRI in place of EGFP. FFR1(GPR40)-Tango (Addgene #66280), FFR2(GPR43)-Tango (Addgene #66281), FFR3 (GPR41)-Tango (Addgene #66282), GPR84-Tango plasmid (Addgene #66377), and GPR120-Tango plasmid (Addgene #66310) were purchased from Addgene. For the GPR120-Rluc plasmid construction, Rluc with Xba I and EcoR I restriction sites was amplified using KOR-link-Rluc vector as the template, and finally linked to GPR120 in pLJM1-GPR120 with a short TCTAGAGGGGGG linker.

## RNA isolation and quantitative real-time PCR
Total RNA from cells or tissues was isolated using the PureLink RNA kit (Invitrogen). 2 µg of RNA was reverse transcribed using the High-Capacity cDNA Reverse Transcription kit (Applied Biosystems #4368814). qPCR was performed using the PowerUP SYBR Green reagent (Applied Biosystems #A25742) in a StepOnePlus Real-Time PCR System (Applied Biosystems). The qPCR reaction maintained temperatures of 50 °C for 5 min, 95 °C for 5 min, 40 cycles of 95 °C for 5 s and 60 °C for 18 s. Ribosomal protein L32 was used as the reference gene. All primers were pre-validated and their sequences are listed in Supplementary Table 1.

## UCP1 immunohistochemistry
iWAT paraffin-embedded sections were deparaffinized and treated with an antigen retrieval process. After blocking with horse serum, the tissues were incubated with a UCP1 primary antibody (1:200) for 1 h. Then, they were washed and incubated with a secondary antibody for 1 h. After washing, the sections were stained with the Vectastain Universal Elite ABC kit (Vector Labs), mounted, and the images were recorded with a Leica DMI4000B fluorescence microscope.

## Immunofluorescence staining
Cells were fixed with 10% buffered formalin and permeabilized with 0.02% Triton X-100. After washing with PBS, 5% BSA in PBS was used to block the cells. After blocking, cells were incubated with primary antibodies anti-UCP1(1:100), anti-COX4(1:100) or anti-c-Myc (1:100), followed by incubation with corresponding secondary antibodies goat anti-rabbit 594 (1:500) or goat anti-mouse 488 (1:500). The images were captured using a Nikon A1Rsi Confocal Microscope.

## Western blot analysis
Liver and iWAT tissues or cells were lysed with RIPA lysis buffer, followed by a 10-min centrifugation at 12,000 × g. Protein concentration was measured using a BCA assay kit (Thermo Fisher Scientific). After adding the sample buffer, protein samples were boiled for 5 min. For western blotting, 25 µg protein samples were loaded for separation using SDS-PAGE and then transferred to nitrocellulose membranes. After blocking, membranes were incubated with the following primary antibodies anti-actin (1:1000), anti-AKT (1:1000), anti-Phospho-Akt (1:1000), anti-Tubulin (1:1000), anti-COX4 (1:1000), anti-Acox1 (1:1000), anti-UCP1 (1:1000), Total OXPHOS Rodent WB Antibody Cocktail (1:400), anti-Tom20 (1:1000), anti-PEX16 (1:1000), anti-PEX5 (1:1000), Anti-PMP70 (1:1000), anti-FASN (1:5000), anti-perilipin-1 (1:1000) or anti-FABP4 (1:1000). The membranes were washed and followed by incubation with corresponding secondary antibodies. The blots were visualized using a chemiluminescent substrate from Thermo Fisher Scientific.

## Adipose tissue histology staining

For H&E staining of tissue slices, fixed adipose tissues were embedded in paraffin, sliced, and stained with H&E at the Musculoskeletal Histology and Morphometry Core at Washington University School of Medicine. After capturing images, adipocyte size was quantified using Adiposoft in ImageJ.

Staining of macrophage markers in whole mounted adipose tissue staining was performed as previously described[47]. Fresh gonadal white adipose tissue was cut into small pieces, approximately 3 mm × 4 mm × 2 mm. After washing with PBS, tissues were incubated with Alexa fluor 647-conjugated F4/80 antibody (1:100, Biolegend #123121) overnight. After antibody incubation, samples were washed and incubated with LipidTox (1:100, Invitrogen #H34475) and DAPI (1:10000) for 2 h. Then samples were washed and mounted, images were taken using a Nikon A1Rsi Confocal Microscope.

## Lipid droplet staining in adipocytes

For Oil Red O staining, cells were fixed with a 4% paraformaldehyde solution for 1 h, washed with isopropanol 2 times, and incubated with Oil Red O solution for 10 min. For visualization of lipid droplets under fluorescence microscopy, fixed cells were stained with LipidTOX according to the manufacturer's instructions.

## Liver RNA-seq analysis

RNA-seq libraries were indexed and pooled according to the library kit manufacturer's protocol. High-throughput sequencing was performed on the HiSeq2500 at the Genome Technology Access Center (GTAC) at Washington University School of Medicine in St. Louis. RNA-seq reads were aligned to the Ensembl release 76 primary assembly with STAR version 2.5.1a[48]. Gene counts were derived from the number of uniquely aligned unambiguous reads by Subread:featureCount version 1.4.6-p5[49]. Isoform expression of known Ensembl transcripts was estimated with Salmon version 0.8.2[50]. All gene counts were then imported into the R/Bioconductor package EdgeR[51] and TMM normalization size factors were calculated to adjust for samples for differences in library size. The TMM size factors and the matrix of counts were then imported into the R/Bioconductor package Limma[52]. Weighted likelihoods based on the observed mean-variance relationship of every gene and sample were then calculated for all samples with the voomWithQualityWeights[53]. Differential expression analysis was then performed to analyze for the differences between conditions and the results were filtered for only those genes with Benjamini-Hochberg false-discovery rate adjusted $p$-values less than or equal to 0.05. For each contrast extracted with Limma, global perturbations in known Gene Ontology (GO) terms, MSigDb, and KEGG pathways were detected using the R/Bioconductor package GAGE[54] to test for changes in expression of the reported log 2 fold-changes reported by Limma in each term versus the background log 2 fold-changes of all genes found outside the respective term. Data have been deposited into the NCBI GEO under the accession number GSE227896.

## Extraction and analysis of fatty acids

The analysis of the total fatty acids released from the liver and serum after acid hydrolysis was according to the protocol previously described[55]. Briefly, to the liver homogenate containing 50 μg protein or 10 μL serum in a glass centrifuge tube, 25 μL 2 mM docosanoic-22, 22, 22-D$_3$ acid internal standard were added. After mixing, 1 mL CH$_3$CN/ 37% HCl (4/1, v/v) was added to each tube, vortexed at full speed for 1 min, and heated at 90 °C for 1 h. Thereafter, the tubes were cooled to room temperature, 2 mL hexane was added, vortexed at full speed for 10 s, and centrifuged at 1740 x g at room temperature for 1 min. The top organic layer was then transferred to a new vial, tried under a stream of nitrogen, reconstituted to 1 mL methanol with 0.25% NH$_4$OH, and loop injected (10 uL) to a Thermo Fisher LTQ Orbitrap Velos for high-resolution electrospray ionization mass spectrometric analysis.

FFA samples were analyzed in the negative-ion mode by manually injecting into ESI source with a 15 μL/min flow of methanol with 0.5% NH4OH. MS$^n$ ($n$ = 2 and 3) spectra were acquired for structural identification, utilizing optimized relative collision energy (30–45%), an activation q value of 0.25, and an activation time of 10 ms to leave a residual precursor ion abundance around 20%. The precursor ion selection window was set to 1 Da for collision-induced dissociation for the high-resolution accurate mass detection in the Orbitrap mass analyzer. Mass spectra were accumulated in profile mode. Data were processed with Thermo Xcalibur software, and offline recalibration was performed using an internal ion of known m/z for accurate mass measurements.

## Serum lipidomic analysis

Global lipidomic analysis was performed on serum from 5 control and 5 *Acox1*-LKO mice by the Metabolomics Innovation Centre (TMIC). The extraction was performed strictly following a modified Folch liquid-liquid extraction protocol. Each aliquot of 6.0 μL of the sample was mixed with an internal standard mixture composed of 15 deuterated lipids, dichloromethane and methanol. A clean-up step was performed with water. Samples were equilibrated at room temperature for 10 min and centrifuged at 16,000 g for 10 min at 4 °C. An aliquot of the organic layer was evaporated to dryness with a nitrogen blowdown evaporator. LC-MS analysis was performed using a Thermo Fisher Dionex UltiMate 3000 UHPLC linked to Bruker Impact II QTOF Mass Spectrometer. The column was a Waters ACQUITY Premier CSH C18 column (100 × 2.1 mm, 1.7 μm particle size), and the column oven temperature was set to 42 °C. Chromatography was carried out using solvent A (10 mM NH4COOH in 50:40:10 MeOH/ACN/Water) and solvent B (10 mM NH4COOH in 95:5 IPA/Water), with the gradient setting as follows: t = 0 min, 5% B; t = 10 min, 40% B; t = 18.8 min, 98% B; t = 20.5 min, 98% B. The flow rate was 250 μL/min. The mass spectrometer was operated in auto MS/MS scan mode to acquire both MS and MS/MS data for all samples. The acquisition rate was 1.44 Hz for MS acquisition and 4–10 Hz for MS/MS spectra acquisition, with an m/z range from 150 to 1500. MS/MS collision energies ramping from 10 eV to 70 eV were used for both positive and negative ionization. Lipid features were extracted and aligned using NovaMT LipidScreener 1.1.0 (Nova Medical Testing Inc.), and the data acquired in positive and negative ionization from each sample experiment were combined. A three-tier identification approach based on MS/MS spectral similarity, retention time and accurate mass match was employed for lipid identification. A nine-tier filtering and scoring approach embedded in NovaMT LipidScreener was employed to calculate MS/MS match scores, restrict the number of matches, and select the best identification. Identified features were normalized by internal standards and statistical analysis was performed in MetaboAnalyst 4.0 (https://www.metaboanalyst.ca/). The pathway enrichment analysis was performed in MetaboAnalyst through input of the compound names of 313 changed lipids which matched to the MetaboAnalyst database.

## Elisa analysis of insulin, TNFα and IL-1β

All ELISA kits used in this project were purchased from Crystal Chem (Downers Grove, IL). Measurements were processed following the manufacturer's protocols.

## Serum TAG assay

Serum TAG levels were measured using the Triglycerides Reagent (Thermo Scientific, #TR22421) and Triglyceride Standard (Pointe Scientific, #T7531STD). In brief, 2.5 μl of the standard or serum sample and 250 μl of the reagent were added to each microplate well. After incubating at 37 °C for 10 min, absorbance at 500 nm was measured using a Tecan plate reader.

For fecal TAG, feces were collected and homogenized in H$_2$O (1/ 24, w/v). Chloroform/methanol (2/1, v/v) was then added to extract

TAG. 300ul of the bottom chloroform layer containing TAG was transferred to a new tube and dried under nitrogen. The TAG content was measured using triglycerides reagent as described above.

## Oxygen consumption rate measurement using Seahorse
iWAT SVF cells from WT mice were seeded in a 24-well seahorse well culture plate, grown to confluence, and induced to become beige adipocytes. Treatments were performed after 3 days of induction. For fatty acid treatment, the cells were treated with vehicle or THA at 40 μM in a maintenance medium for 6 days. Lenti-shGPR120 was used for the knockdown of GPR120 with Lenti-shScramble used as control. For hWAT, differentiation of confluent cells was induced in a 24-well Seahorse cell culture plate for 9 days, then the cells were treated with vehicle or THA at 25 μM in a differentiation medium for another 9 days. After the treatment, the OCR of mice iWAT and hWAT was measured at baseline or after injection of oligomycin, FCCP, and antimycin/rotenone.

## Mitochondrial respiration of frozen tissue
Mitochondrial respiration of frozen iWAT tissue was measured as previously described[56]. Briefly, frozen iWAT samples were thawed on ice and homogenized using a Glass Dounce Tissue Grinders in MAS buffer (70 mM sucrose, 220 mM mannitol, 5 mM KH2PO4, 5 mM MgCl2, 1 mM EGTA, 2 mM HEPES pH 7.4). Then, homogenates were centrifuged at 1000 $g$ for 10 min at 4 °C, and the supernatant was collected. Protein concentration was determined using a BCA kit (ThermoFisher). 25 μg protein was loaded into each well of an XF24 cell culture microplate. Substrate injections were as follows: 5 mM succinate + 2 μM rotenone for port A, 2 μM rotenone + 4 μM antimycin for port B, 0.5 mM TMPD + 1 mM ascorbic acid for port C, and 50 mM azide for port D.

## mtDNA copy number
Total genomic DNA from tissues or cells was isolated using the DNeasy Blood & Tissue Kit (QIAGEN). The mtDNA copy number was measured as described previously[42]. The primers used for mitochondrial and nuclear DNA were as follows: mitoDNA-F TTAAGACA CCTTGCCTAGCCACAC; mitoDNA-R CGGTGGCTGGCACGAAATT; NucDNA-F ATGACGATATCGCTGCGCTG; NucDNA-R TCACTTAC CTGGTGCCTAGGGC. qPCR was performed using 50 ng of total DNA. The mtDNA was normalized to the nuclear DNA.

## Indirect calorimetry and body composition analysis
VO2, VCO2, and RER were measured by indirect calorimetry using a PhenoMaster (TSE Systems) metabolic cage system as described previously[42]. Mice were acclimated in system cage for 4 h, then data were collected for at least 24 h for each mouse and analyzed using CalR[57]. For CL 316,243 induced OCR, mice were firstly acclimated to the system for 3 h, CL 316,243 was intraperitoneal injected at dose of 1 mg/kg, then data were further collected for 1 h. Fat and lean mass were measured using an EchoMRI system.

## Ca²⁺ staining and assay
293T cells expressing mouse GPR120 were seeded in a black clear-bottom 96-well plate. The cells were treated with vehicle, PA, EPA, THA, or GSK137647A at the indicated concentration for 16 h. The cells were then incubated in Fluo-4 Direct calcium reagent solution (Thermo Fisher #F10471) for 1 h. After that, the fluorescence was read, and images were captured using a Leica microscope.

## GPCR Tango assay
Tango assay was performed essentially as previously described[32]. Briefly, HTLA cells transfected with a specific GPCR-Tango plasmid were seeded in a poly-l-lysine coated plate and cultured in starvation medium (DMEM supplemented with 1% dialyzed FBS and 1% penicillin/

streptomycin). Then, the cells were treated with various fatty acids at different concentrations. After an 18-h treatment, the cell media was replaced with luciferase assay reagent (Promega Bright-Glo™ Luciferase Assay System) and incubated for 10 min at room temperature. After incubation, luminescence was immediately measured with a luminescence plate reader from Tecan.

## BRET assay
HEK-293T cells were transfected with a 1:5 DNA ratio of GPR120 engineered to fuse with Renilla luciferase (RLuc8) at the C-terminus and β-arrestin 1 ligated to mVenus at the N-terminus in the presence of Transit 2020. 24 h post-transfection, the transfected cells were seeded in a poly-l-lysine coated 96 well plate at a density of 50,000 cells per 200 μl per well and cultured in starvation medium (DMEM supplemented with 1% dialyzed FBS and 1% penicillin/streptomycin). Next day, 60 μL per well of the RLuc substrate (5 μM coelenterazine h) was added to the cells and incubated for 5 min in the dark at room temperature to activate arrestin after washing the cells with 60 μL per well of a drug buffer (1× HBSS and 20 mM HEPES, pH 7.4) per well. Following this, cells were treated with 30 μL per well of the (3×) various fatty acids at different concentrations that were prepared in buffer (1× HBSS and 20 mM HEPES, pH 7.4) supplemented with 0.3% BSA and incubated for another 5 min. A Mithras LB940 multimode microplate reader was immediately used to measure the BRET ratio for β-arrestin 1 recruitment. This was done by detecting the ratio of mVenus emission at 485 nm to Rluc emission at 530 nm for 1 s per well. The potency of the fatty acids was calculated using GraphPad Prism 9 software after plotting and the net BRET ratios against various fatty acids concentrations.

## Statistics
Data are reported as mean ± standard error of the mean (SEM) unless stated otherwise. Statistical comparisons between two groups were performed by using unpaired $t$ test. Analysis of variance (ANOVA) was used for more than 2 groups. A $P$ value less than 0.05 was considered statistically significant. Statistical analysis and graphs were generated using GraphPad Prism software.

## Reporting summary
Further information on research design is available in the Nature Portfolio Reporting Summary linked to this article.

## Data availability
RNA-sequencing data presented in Fig. 3 have been deposited into the NCBI GEO under the accession number GSE227896. Data for supplementary Fig. 1a were extracted from the GEO database: GSE61260. The serum lipidomic data are available in the Metabolomics Workbench database[58] under accession number ST003142. All other data supporting this study are available within this Article and Supplemental Information. The source data generated in this study have been deposited in the Figshare database (https://doi.org/10.6084/m9.figshare.25521334). Source data are provided with this paper.

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

## Acknowledgements

This work was supported by a grant from the Washington University–Centene Corporation Personalized Medicine Initiative and by NIH grants DK115867, DK118333, and DK132239. The authors also acknowledge core services of the Washington University Diabetes Research Center (DK020579), Mass Spectrometry Resource (GM103422), and the Nutrition Obesity Research Center (DK056341).

## Author contributions

D.L. and A.H. designed and conducted experiments and performed data analysis. M.T., M.M., A.E.D., D.H., and X.L. performed experiments. B.K. performed data analysis. F-F.H. performed mass spectrometry analysis. T.C. and C.F.S. designed experiments and provided reagents. M.B. designed experiments and performed data analysis. I.J.L. conceived the study, designed experiments, and performed data analysis. D.L. and I.J.L. wrote the paper. All authors read the manuscript and provided comments.

## Competing interests

The authors declare no competing interests.
