## [Peer Review File · Nature Communications]

Liver ACOX1 Regulates Levels of Circulating Lipids that Promote Metabolic Health through Adipose RemodelingREVIEWER COMMENTS

Reviewer #1 (Remarks to the Author):

The manuscript by Lu and coworkers entitled "Liver Acox1 Regulates Levels of Circulating Lipids that Promote Metabolic Health through GPR120" is of great relevance and high interest demonstrating for the first time a role of peroxisomal functions in a liver-adipose signaling axis. The molecular mechanism is analyzed in great detail with very convincing experiments. Many findings are confirmed by further confirmatory experiments. THA the endogenous anabolic precursor in *de novo* DHA synthesis mediates the effect if the DHA synthesis is blocked and the precursor accumulates. The only point that I think would be nice to be targeted is to exclude the effect also of other substrates that accumulate under these conditions. I do understand that in the last figure it is nicely demonstrated that THA can mediate this effect. Still it would be nice to show in very view key experiments that saturated VLCFAs (e.g. C26:0) is not able to mediate the same effect.

Comments:

1. Introduction: all known substrates of peroxisomal β -oxidation should be named including DHA, leukotrienes, prostaglandins and branched chain FA.
2. The role of Acox2 should be discussed with regard to the substrate specificity (Fig. 1A can still stay without other enzymes of β -ox).
3. It should be discussed that DHA need to undergo one round of peroxisomal β -oxidation in order to be *de novo* synthesized if this does not happen the precursor C24:6 n-3 accumulates and after one more elongation C26:6 n-3 is formed. Thus the appearance of these FAs in the context of peroxisomal β -oxidation deficiency is expected and not surprising.
4. For me Fig 1N appear very similar between Control and Acox1-LKO thus the sentence: "These data suggested marked differences between control and Acox1-LKO mice pointing towards phenotypic remodeling in macrophage population." seems not appropriated at this position.
5. Fig 2B and Extended 2F: it is interesting to see that the differences in gene expression is more extended in the normal chow diet than in HFD – does this not come as a surprise? It is worth to mention in a half sentence before indicating the there are no changes in the classical BAT.
6. Fig 2H here not the delipidated serum but also the serum of wt mice should be used as a control to demonstrate that the effect is derived from the ACOX1-LKO!
7. Fig 3 H cannot be interpreted as it is totally unclear whether a changed lipid is up or downregulated and thus may or may not be in agreement with increased or downregulated genes. Fig 3D just show that many will be up and many will be downregulated of the indicated lipids. This can go into the supplementary figures but a volcano blot with an additional naming of the top 10 lipids upregulated and top 10 lipids downregulated would be more informative. Very likely VLCFA containing lipids will be the top up. It is unclear why C24:0 and C26:0 is not depicted in Extended Fig D3 – it would be expected to be among the highly changed lipids.
8. It would be very interesting whether or not saturated VLCFAs C26:0 do have a agonistic effect on GPR120. This would be an important experiment as it is known to be a major substrate for Acox1 and does also accumulate next to the DHA precursor.

Reviewer #2 (Remarks to the Author):

In this work by Lu and colleagues the authors suggest that a liver-specific knockout of *Acox1*- the rate-limiting enzyme in peroxisome beta oxidation-results in increased levels of omega 3 very long chain fatty acids (VLCFAs). Through their release to the circulation and activation of GPR120, the authors suggest that these VLCFAs induce a browning phenotype that improves the metabolic phenotype of the mice. This notion is based on both genetic knockout models and treatments with tetracosahexaenoic acid (THA) in vivo and studies in cell models. Overall, the work has some merits but the reported effects are rather small and several issues are unclear. My main concerns (not necessarily in order of importance) are:

Overall comments:

1. What are the clinical implications of this work? Do the authors have any data suggesting for instance that variations in liver ACOX1 expression is linked to a healthier phenotype? Other genetic support, eQTLs? A quick search on the internet suggests that ACOX1 deficiency results in VLCFA accumulation and very severe disorders in humans (e.g. pseudo-neonatal adrenoleukodystrophy, see PMID2894756 and multiple other papers)-is this really a druggable target?
2. The assessments in mice appear to be performed in male mice, although this is only clarified in one figure (Figure 6A). Have the authors performed any analyses in female mice?

Specific comments

3. In figure 1G, the browning phenotype in iWAT appears to be very mild as there is no obvious increase in multilocular adipocytes. In fact, the reduction in fat cell size appears proportional to the reduction iWAT mass (Figure 1E). Also, iWAT is a fairly small depot, can the reduction of this depot explain the reduction in total fat mass (approximately 5 grams) shown in panel 1D?
4. There is a mix of studies in iWAT and gWAT, despite the fact that the authors focus on adipocyte browning which only occurs in the former depot. Were assessments of pAKT/AKT done in iWAT (and not only liver and gWAT shown in Figure 1L and 1M)? Also, calculations of pAKT/AKT ratios derived from the images should be provided in these panels.
5. The focus on gWAT inflammation Figure 1N-S is not really clear. Do the authors hypothesize that this effect also contributes to the improvement in glucose tolerance? Is this also mediated via VLCFAs (given that the mass and morphology of gWAT is not altered in the *Acox1* KO mice).
6. In Figure 2B, the browning effect is only really detectable in chow diet fed mice. This appears clear from ED Figure 2F (displaying the HFD animals), where only a fairly small and variable effect on *Ucp1* mRNA expression is observed. This argues again for a very mild browning effect that is almost entirely abrogated by HFD.
7. In general, the western blots only display protein levels of UCP1 where the fold-changes are rather small and somewhat variable, what about the abundance of other OXPHOS proteins?
8. The studies in Figure 4 were done using the same concentration (50 mikrom) for EPA, THA and HHA. How was this concentration chosen? Did the authors perform any titration experiments? Data in Figure 6 suggest that THA levels are around 2 mikrom in wildtype mice and 15 mikrom in the *Acox1* KO.
9. In ED Figure 4, how was the expression of white adipocyte marker genes affected? For example: LIPE, PNPLA2, PLIN1 and ADIPOQ.
10. The data in Figure 5G show a rather small effect of THA on UCP1 mRNA (40% or so) and protein levels (not quantified). In Figure 4D the effect on *Ucp1* mRNA is around 250%. This is still a very low effect as browning usually induces >10-100-fold increases in *Ucp1* gene expression in white adipocytes.
11. The weakest part are the data in Figure 6. Treatment with THA results in small increases in serum-THA (Figure 6B) without any effects on body weight, body composition or glucose tolerance (despite the authors stating that there is an effect, all p-values are >0.05) (Figure 6D-F). In line with this, there are no significant increases in browning gene markers (Figure 6I) except for UCP1 which would be non-significant if correction for multiple comparisons would have been done. I don't think the data in this Figure strengthens their hypothesis.

Reviewer #3 (Remarks to the Author):

This is a well-conducted study that provides novel insights into the role of liver Acox1 in regulating circulating lipid levels, in particular omega-3 VLCFAs and their impact on metabolic health, in particular on the browning of adipose tissue. In vitro experiments indicate that these protective effects of THA are mediated by GPR120 in adipocytes, but in vivo evidence for this mechanism is lacking. In general, the claims of this study are well supported by data, experiments are technically sound, with appropriate controls, and presented in sufficient detail. However, there are several aspects that require some additional experiments and modifications that need to be done in the manuscript to improve clarity. Once these issues are addressed I think this manuscript will advance our understanding of the role of peroxisomal fatty acid oxidation in metabolic homeostasis, and provide more insights about intricate interorgan metabolic connections. Here are my comments.

Major comments

1. One of my main concerns is the lack of data regarding the specific inhibitory effects of TDYA on Acox1 and peroxisomal beta-oxidation. In other words, how are the authors sure that TDYA does not have off-target effects on other enzymes? At least they should add data of classical VLCFA levels such as C26 levels in mice treated with TDYA. Moreover, there should be a discussion about the appearance of differences in BW gain much later than in Acox1-LKO mice. Why not represent food intake in grams as it is usually done? This representation in % of BW is confusing and not informative.

But having said all that, without definite proof for a specific inhibition of ACOX1 by TDYA my recommendation is to withdraw these data from the manuscript, which is still very robust even without these data, but its inclusion may confound future research on peroxisomal b-ox inhibition, until we know about the off-target effects of TDYA.

2. The authors claim that the omega-3 VLCFAs they identified are 'previously uncharacterized VLCFA' However, this statement is misleading, as these fatty acids have been known to be metabolized in peroxisomes for decades, at least for THA. Indeed, patients with peroxisomal disorders frequently present deficits in C22:6 (DHA). It is also known that they accumulate in global Acox1 KO mice. While the specific omega-3 VLCFAs they identified may not have been studied in the context of GPR120 activation and metabolic health, their existence and their connection with peroxisomal metabolism are well established. Therefore, this fact should be acknowledged (PMID 8847474, PMID 11734571 and PMID 11855929). The original reference of global Acox1 KO mice that show extensive changes in the liver should be cited too (PMID 8798738).

3. Given the importance of peroxisomal metabolism in the liver in this manuscript, it will be a great addition to this paper to evaluate peroxisomal number/proliferation in the livers of Acox1-LKO mice, and their changes with HFD. That could help understand the role of hepatic peroxisomes in diet-induced obesity.

4. The characterization of browning of white adipocytes induced by serum of Acox1 LKO should be further expanded besides mRNA levels of browning genes, in order to evaluate whether the concentrations present in the Acox1-LKO serum exert similar effects as the ones seen by THA treatment. Thus, repeating the experiments shown in Figure 4 for THA (or at least some of them) with the serum from Acox1-LKO mice will be very informative.

5. All the claims related to the role of GPR120 in this liver-adipose axis need to be toned down across the manuscript including the title, since authors do not provide in vivo evidence for this role of GPR120. Their experiments in vitro are sound and relevant but not enough to place GPR120 in this axis that require in vivo communication between these two organs. GPR120 expression could be measured in vivo (chow and HFD in control and Acox1-LKO to get some clues about changes in this GPCR in vivo). Following up on this last comment, authors assume that the decrease in circulating fatty acids is caused by changes in FAO, but it could be also caused by changes in lipolysis. Considering that agonists of GPR120 have a profound effect on lipolysis (PMID

33091626), this is an aspect that should be investigated in Acox1-LKO mice (chow and HFD conditions).

6. The use of unpaired t-test to evaluate differences in multiple figures (Figure 4, Figure 6, Ext. Data Fig. 1A-C, and more) does not seem correct as more than two groups are being compared. Please revise the statistical analysis across the manuscript and use the appropriate statistical test to evaluate those differences.

Minor comments

1. In Fig.1, at what point were peroxisomal enzymes expression measured? Only mention in the methods of duration of HFD is 9 weeks before THA treatment, but Fig. 1C shows 11 weeks of HFD treatment. It is not clear which time point was the one used to measure mRNA. Please clarify. Did authors measure the expression of Acaa1a or Acaa1b? Why n=3 only for gene expression, while in Fig. 1S the n is much higher?

2. The way that the changes in leukocytes in SVF are shown in Ext. Data 1E is confusing. In the text the authors talk about total numbers of SVF cells but they represent per g of fat. Perhaps it would be more visual to show the total number of cells, so it would be clear that there are no differences despite the increase in gWAT weight. Alternatively, the text could be changed to better reflect what has been plotted.

3. EE data. It is not clear in the text whether EE was measured in HFD-fed control and Acox1-LKO mice or in chow-fed. Assuming it was done in HFD-fed mice, body masses are different between control and Acox1-LKO mice. Thus, a regression-based analysis of EE using body mass as a covariate should be included in the analysis of EE. Data looks convincing but that additional analysis would give more information. Indeed authors did this in Ext Data Fig. 6 with other conditions, so why not doing the same here?

In that same data from metabolic chambers in Ext. Data Fig. 2, 5 g of HFD per day seems quite high for B6 mice. Do authors have food intake data (ideally cumulative) from the same mice shown in Fig.1, where progression of BW is shown?

4. Fig 2D shows UCP1 IHC in Control and Acox1-LKO. In how many mice was this experiment performed? Information is not available in figure legend. Why was mitochondrial respiration performed in frozen tissue when this is suboptimal in comparison with fresh tissue? Did authors observe more mitochondrial biogenesis in iWAT of Acox1-LKO mice?

5. In the reporting of changes in genes involved in omega-oxidation it would be useful for the reader to have references to a review discussing the omega-oxidation pathway (either PMID 37140888 or PMID 24126285).

6. Add units for scale in Fig. 3B (z-score I assume).

7. If authors have acylcarnitine data from the serum/hepatic lipidome I suggest they include them in a separate graph/s, as it would help to understand the impact of FAO in these Acox1-LKO mice, and more importantly since these metabolites have been shown to have a thermogenic role.

8. Quantification is missing for WB in Fig. 5H

9. Figure 1A depicts peroxisomal beta-oxidation pathway with the peroxisomal transporter ABCD1. Considering the focus on C24:6n-3 (THA), ABCD2 should be added to that schema, as it is the hypothesized transporter for THA, and there it no mention to it across the manuscript (PMID 21145416 and PMID 18854420).

- Other comments:

- Intro: Following up on my previous comments regarding omega-3 VLCFA and peroxisomes, authors should change the introduction to acknowledge this previous knowledge, and also include branched-chain fatty acids in the substrates metabolized by peroxisomes. I would also suggest

adding to the intro that VLCFA beta-oxidation is severely reduced in the liver of ACOX1-LKO mice, so the reader is aware of that before diving into the results where it is mentioned that peroxisomal beta-ox is dysfunctional in ACOX1-LKO.

- Methods/Reporting:

Please provide more information about the analysis of the metabolome with Metaboanalyst to improve reproducibility of the findings. How were the metabolites imputed in Metaboanalyst (KEGG ids or HMDB ids, and how many were used) to perform the pathway enrichment?

How were TG measured?

Please report the total number of animals used for this study. There is no mention regarding the sex of the animals used in this study.

The inclusion of the tissue used for measurements in the figure panels would facilitate the comprehension of these large figures

Please provide details regarding the antibodies used in this study and full blots of Western Blots with MW markers.

- Writing: Make a consistent use of beta/ β latin and greek nomenclature. p6, L174-175: ... circumstances that resulting in the... (result?)

- Discussion:

Authors claim that "Whether peroxisomal β -oxidation influences systemic metabolism to affect diet-induced obesity has not been investigated". That is an incorrect statement. Zeng et al. did that using TDYA, the inhibitor authors used in this manuscript. Therefore, this sentence should be changed to reflect a more specific claim about the authors' findings.

Acox1 inactivation and hepatomegaly. I think this is an important aspect that has not been properly discussed. What is the hypothesis of the authors for this discrepancy between global KO (even Acox1Lampe) and Acox1-LKO mice, even more when RNA-seq data show activation of PPAR signaling pathway and multiple PPAR-alpha targets, that in mice drive hepatomegaly.

RE: "Liver ACOX1 Regulates Levels of Circulating Lipids that Promote Metabolic Health through Adipose Remodeling" by Lu et al.

Manuscript No.: NCOMMS-23-39474-T

We thank the Reviewers for their insightful comments and feedback, which have enabled us to enhance the quality of our manuscript. We have carefully reviewed and addressed all the concerns and suggestions raised by the three reviewers and have incorporated new experiments and revised the text accordingly. Major changes in the revised manuscript are highlighted in red. As a result of these changes, we believe that this is a much improved manuscript. Our point-by-point responses to the reviewers' comments are provided below.

Reviewer #1 (Remarks to the Author):

The manuscript by Lu and coworkers entitled "Liver Acox1 Regulates Levels of Circulating Lipids that Promote Metabolic Health through GPR120" is of great relevance and high interest demonstrating for the first time a role of peroxisomal functions in a liver-adipose signaling axis. The molecular mechanism is analyzed in great detail with very convincing experiments. Many findings are confirmed by further confirmatory experiments. THA the endogenous anabolic precursor in *de novo* DHA synthesis mediates the effect if the DHA synthesis is blocked and the precursor accumulates. The only point that I think would be nice to be targeted is to exclude the effect also of other substrates that accumulate under these conditions. I do understand that in the last figure it is nicely demonstrated that THA can mediate this effect. Still it would be nice to show in very view key experiments that saturated VLCFAs (e.g. C26:0) is not able to mediate the same effect.

Response: Thank you for your comments and thoughtful input into our paper. We have performed a considerable amount of additional work according to your and other reviewers' suggestions, which has greatly improved this manuscript. Point-by-point responses to your comments are provided below.

Comments:

1. Introduction: all known substrates of peroxisomal β -oxidation should be named including DHA, leukotrienes, prostaglandins and branched chain FA.

Response: Thanks for this suggestion. We have revised the Introduction to include the following text on page 2:

Mitochondrial β -oxidation plays a more important role in catabolism of short-, medium-, and most long-chain fatty acids, while peroxisomal β -oxidation is required for degradation of very-long-chain fatty acids (VLCFAs; ≥ 22 carbons), di- and trihydroxycholestanic acid (DHCA and THCA), long chain dicarboxylic acids, 2-methyl branched chain fatty acids (BCFA), leukotrienes and prostaglandins^{4,5}

2. The role of Acox2 should be discussed with regard to the substrate specificity (Fig. 1A can still stay without other enzymes of β -ox).

Response: We have revised the introduction to include a brief discussion of substrate specificities of all three family members of acyl coA oxidase family of proteins, including Acox2:

“ACOX1 is specific to the degradation of straight chain saturated and unsaturated VLCFAs. ACOX2 promotes β -oxidation of 2-methyl-BCFA, such as pristanic acid, and bile acid intermediates, including DHCA and THCA. ACOX3 is also involved in catabolism of 2-methyl branched fatty acids, suggesting that it has a partially overlapping function with ACOX2 ⁶.”

3. It should be discussed that DHA need to undergo one round of peroxisomal β -oxidation in order to be de novo synthesized if this does not happen the precursor C24:6 n-3 accumulates and after one more elongation C26:6 n-3 is formed. Thus the appearance of these FAs in the context of peroxisomal β -oxidation deficiency is expected and not surprising.

Response: We appreciate your comments. We have added the following text on page 2 of the revised manuscript to discuss the role of C24:6 in DHA synthesis:

“Through its role in oxidation of certain polyunsaturated fatty acids, ACOX1-mediated β -oxidation affects the levels of the omega-3 (ω -3) fatty acids eicosapentaenoic acid (EPA; C20:5) and docosahexaenoic acid (DHA; C22:6). In mammals, peroxisomal β -oxidation not only oxidizes DHA, but also affects production of DHA through a process involving desaturation and elongation of α -linolenic acid (ALA, 18:3-3) to generate tetracosahexaenoic acid (THA; C24:6), which is then chain-shortened to DHA by peroxisomal β -oxidation^{7, 8}. Peroxisome β -oxidation deficiency results in decreased levels of DHA and accumulation of longer chain fatty acids⁹.”

Although accumulation of C24:6 through inhibition of DHA synthesis is expected, the release of C24:6 and related longer chain fatty acids into the circulation is surprising. Moreover, the preferential accumulation of these polyunsaturated VLCFAs as opposed to saturated VLCFAs, such as C24:0 and C26:0 is unexpected. Our results indicate that a combination of impaired ACOX1-mediated β -oxidation itself and the associated gene expression changes, such as increased expression of desaturases, elongases, and fatty acid transporters contribute to the observed circulating lipid profile in ACOX1-LKO mice.

4. For me Fig 1N appear very similar between Control and Acox1-LKO thus the sentence: “These data suggested marked differences between control and Acox1-LKO mice pointing towards phenotypic remodeling in macrophage population.” seems not appropriated at this position.

Response: We repeated this experiment with a larger cohort of mice and obtained consistent results. The updated results are presented in revised Fig. 1n-s. Using unbiased clustering analysis of CD45+ cells we have identified 12 distinct immune cell types, among them adipose tissue macrophages (ATMs) (Fig. 1n). Notably, this analysis revealed a CD9+ ATM subset, recognized for its association with pathological obesity¹⁷ (Supplementary Fig. 1l). Quantification of per gram cell numbers showed normal number of various immune cells (Supplementary Fig. 1m-o), except ATMs, which were significantly decreased in Acox1-LKO (Fig. 1o), and this decrease was mostly driven by lowered numbers of CD9+ ATMs subset in Acox1-LKO (Fig. 1p). Cell density plots suggested phenotypic remodeling in the ATM population in Acox1-LKO (Fig. 1q). This was accompanied by increased proportions of CD11c-CD206+ macrophages that are known to represent

anti-inflammatory phenotype^{18, 19} (**Fig. 1r**). Simultaneously, there was a decrease in pro-inflammatory macrophages CD206+CD11c+ ATMs which are associated with obesity and insulin resistance²⁰ (**Fig. 1s**). These data suggest decreased adipose tissue inflammatory status in *Acox1*-LKO as indicated by reduced infiltration by inflammatory ATM subsets, consistent with the improved metabolic profile.

5. Fig 2B and Extended 2F: it is interesting to see that the differences in gene expression is more extended in the normal chow diet than in HFD – does this not come as a surprise? It is worth to mention in a half sentence before indicating the there are no changes in the classical BAT.

Response: We have analyzed iWAT browning in *Acox1*-LKO mice under HFD through measurement of gene expression (revised **Supplementary Fig. 2f**) and protein (revised **Supplementary Fig. 2i**) levels with additional cohorts of mice. Although thermogenic gene expression was increased in iWAT of *Acox1*-LKO fed either chow diet or HFD, the magnitude of increase was greater in chow fed animals. This is because high fat feeding dramatically decreases the expression brown gene genes in iWAT, as show below for UCP1 (**Figure A**) and previously reported by others (see for example Hu et al, Nature Comm, 2023, fig. 3b; PMID: 38036537). A brief discussion related browning under chow vs. HFD is included on page 6 of the revised manuscript.

Figure A. UCP1 gene expression in iWAT of WT mice fed chow or high fat diet.

6. Fig 2H here not the delipidated serum but also the serum of wt mice should be used as a control to demonstrate that the effect is derived from the ACOX1-LKO!

Response: Thank you for your suggestion. We repeated this experiment and normalized all treatments to the control serum. Our results (**Fig. 2j**) are consistent with our previous results indicating that *Acox1*-LKO mice serum treatment increases the expression of thermogenic gene UCP1, *Elovl3* and *COX4* compared to control mice serum. Delipidated WT and *Acox1*-LKO mice serum did not promote the expression of browning genes. Overall, our data indicate that the iWAT adipocytes browning in *Acox*-LKO mice is induced by changes in circulating lipids.

7. Fig 3 H cannot be interpreted as it is totally unclear whether a changed lipid is up or downregulated and thus may or may not be in agreement with increased or downregulated genes. Fig 3D just show that many will be up and many will be downregulated of the indicated lipids. This can go into the supplementary figures but a volcano blot with an additional naming of the top 10 lipids upregulated and top 10 lipids downregulated would be more informative. Very likely VLCFA containing lipids will be the top up. It is unclear why C24:0 and C26:0 is not depicted in Extended Fig D3 – it would be expected to be among the highly changed lipids.

Response: In the original **Fig. 3h**, we sought to explain how transcriptional changes and changes in serum lipids contribute to the accumulation of C24:6. This figure has been moved to a new **Supplementary Fig. 3f**. It is noteworthy that all the increased VLCFAs in *Acox1*-LKO mice were unsaturated fatty acids, while

the saturated very long chain fatty acids, including C24:0 and C26:0, were not significantly changed in Acox1-LKO mice (see **Figure B** below), unlike in the Acox1 global deficiency mouse model (PMID: 8798738), where they were increased. The increased levels of polyunsaturated fatty acids, such as C24:6 and C26:6, could be explained by impaired β -oxidation due to the knockout of Acox1, as well altered expression of genes involved in fatty acid elongation and desaturation.

Thank you for the suggestion regarding **Fig. 3d**. We have replaced the heatmap with a volcano plot that identifies top changed lipids. The volcano plot shows that several species of fatty acids with very long chain and multiple double bonds were among the top increased lipids, while the decreased lipids included phospholipids containing DHA (22:6) and fatty acids with less unsaturation. These results are consistent with our original analysis of the lipidomic data.

8. It would be very interesting whether or not saturated VLCFAs C26:0 do have a agonistic effect on GPR120. This would be an important experiment as it is known to be a major substrate for Acox1 and does also accumulate next to the DHA precursor.

Response: The saturated very long chain fatty acids, like C24:0 and C26:0, are not significantly increased in the Acox1-LKO mouse model. However, we agree that it is of interest to see if these saturated VLCFAs

could activate GPR120. Thus, we determined whether C26:0 could activate GPR120 using Tango GPCR assay system. As shown in **Figure C**, C26:0 did not have agonistic effect on GPR120.

Reviewer #2 (Remarks to the Author):

In this work by Lu and colleagues the authors suggest that a liver-specific knockout of Acox1- the rate-limiting enzyme in peroxisome beta oxidation-results in increased levels of omega 3 very long chain fatty acids (VLCFAs). Through their release to the circulation and activation of GPR120, the authors suggest that these VLCFAs induce a browning phenotype that improves the metabolic phenotype of the mice. This

notion is based on both genetic knockout models and treatments with tetracosahexaenoic acid (THA) in vivo and studies in cell models. Overall, the work has some merits but the reported effects are rather small and several issues are unclear. My main concerns (not necessarily in order of importance) are:

Response: Thank you for your comments. We appreciate your recognition of the merits of our work. Our point-by-point responses to your concerns are provided below.

Overall comments:

1. What are the clinical implications of this work? Do the authors have any data suggesting for instance that variations in liver ACOX1 expression is linked to a healthier phenotype? Other genetic support, eQTLs? A quick search on the internet suggests that ACOX1 deficiency results in VLCFA accumulation and very severe disorders in humans (e.g. pseudo-neonatal adrenoleukodystrophy, see PMID2894756 and multiple other papers)-is this really a druggable target?

Response: Although prenatal global knockout of *Acox1* in the 129/Sv mouse strain leads to severe abnormalities, including growth retardation, hepatic steatosis and hepatocellular carcinoma^{10, 40}, our work shows that mice with liver-specific knockout of *Acox1* on the C57BL/6J genetic background are healthy with normal liver function. The phenotypic difference between our model and the global knockout mice might reflect difference in genetic backgrounds, since 129/Sv mice are known for having increased susceptibility for developing cancer⁴¹. It also possible that extrahepatic deletion of *Acox1* in the global knockout mouse model contributes to liver disease. However, our results indicate that systemic inhibition of ACOX1 activity in mice using a pharmacological inhibitor does not lead abnormalities and instead improves metabolic health. Thus, ACOX1 is a potentially druggable target. Our work indicates that liver peroxisomal β -oxidation activity regulates whole body metabolic homeostasis by influencing the circulating levels of omega-3 VLCFAs. Based on this, the specific inhibition of liver peroxisomal β -oxidation function by liver-targeted delivery of *Acox1* antisense oligonucleotide using GalNac conjugates (e.g., PMID: 32592692) or dietary supplementation of these beneficial lipids represent potential strategies to treat obesity-associated metabolic disorders.

We are not aware of any eQTLs associated with obesity. However, search of published datasets show that liver *Acox1* expression is linked to BMI in humans. For example, search of publicly available transcriptome

profiling data (GEO: GSE61260 comparing liver gene expression of 58 lean control and 76 obese subjects, revealed that *Acox1* gene expression level is significantly higher in the livers of obese individuals than in lean controls (new **Supplementary Fig. 1a**). Dataset GSE15653 (**Figure D**) also shows that liver *Acox1* gene expression is nearly significantly higher in obese people compared to lean subjects. In addition, our data in mice (**Fig. 1b**) shows that *Acox1* expression in the liver of HFD-induced obese mice is higher than that in chow diet-fed mice. These data

indicate that liver Acox1 expression is related to metabolic homeostasis in both humans and mice.

2. The assessments in mice appear to be performed in male mice, although this is only clarified in one figure (Figure 6A). Have the authors performed any analyses in female mice?

Response: As noted on page 16 of the revised manuscript, “Metabolic phenotyping experiments were primarily conducted using male animals. In selected experiments where female mice were used, the sex is specified in the figure legend.” Our results indicate that although HFD feeding did not significantly affect the body weight in female Acox1-LKO compared to control animals (new **Supplementary Fig. 1f**), the mutant mice had improved glucose homeostasis (new **Supplementary Fig. 1g**). This suggests that hepatic ACOX1 inactivation has a primary effect on insulin sensitivity, independent of body weight difference. In addition, we report that female ACOX1 KO mice exhibit iWAT browning (**Supplementary Fig. 2g**), consistent with the notion that the metabolic benefits of adipocyte browning extend beyond regulation of energy balance.

Specific comments

3. In figure 1G, the browning phenotype in iWAT appears to be very mild as there is no obvious increase in multilocular adipocytes. In fact, the reduction in fat cell size appears proportional to the reduction iWAT mass (Figure 1E). Also, iWAT is a fairly small depot, can the reduction of this depot explain the reduction in total fat mass (approximately 5 grams) shown in panel 1D?

Response: The magnitude of increase in brown gene expression in our HFD-fed Acox1-LKO mice is comparable to increase previously reported by others where the body weight was significantly decreased (see for example, PMID: 38036537). Although readily seen in iWAT of cold-treated mice, multilocular adipocytes are difficult to see with high fat feeding because thermogenic gene expression is markedly decreased in iWAT of HFD-fed animals (see **Figure A** above).

Nevertheless, our gene expression and Western blot data show that Acox1-LKO mice consistently have higher iWAT browning level than WT mice. It is likely that HFD feeding dampens the iWAT browning in both WT and Acox1-LKO mice. Given that Acox1-LKO mice have increased energy expenditure (**Fig. 2a-b**), decreased body weight (**Fig. 1c**), and improved glucose homeostasis (**Fig. 1i**), the browning in these mice is physiologically relevant.

We changed **Fig. 1e** to show absolute weights of tissues. The data shows that the decrease in iWAT mass accounts for approximately 1 gram reduction in body weight. Subcutaneous WAT is a major contributor to the total fat mass and iWAT fat pad is only one of the depots of subcutaneous fat. We did not collect all subcutaneous fat pads. It is likely that decrease in the remaining subcutaneous WAT accounts for other 4g reduction in body weight observed in HFD-fed Acox1-LKO mice.

4. There is a mix of studies in iWAT and gWAT, despite the fact that the authors focus on adipocyte browning which only occurs in the former depot. Were assessments of pAKT/AKT done in iWAT (and not only liver and gWAT shown in Figure 1L and 1M)? Also, calculations of pAKT/AKT ratios derived from the images should be provided in these panels.

Response: Besides browning of iWAT, our results indicates that Acox1 ablation decreases inflammation in gWAT, a visceral adipose depot that is more susceptible to obesity-associated recruitment of immune cells.

We also assessed the pAKT/AKT by Western blot analysis in iWAT. The Western blot results and calculations of pAKT/AKT are shown in revised **Fig. 1m**. Our results reveal that Acox1-LKO mice have markedly increased insulin-stimulated Akt phosphorylation in iWAT.

5. The focus on gWAT inflammation Figure 1N-S is not really clear. Do the authors hypothesize that this effect also contributes to the improvement in glucose tolerance? Is this also mediated via VLCFAs (given that the mass and morphology of gWAT is not altered in the Acox1 KO mice).

Response: Our results show that although HFD-fed Acox1-LKO female mice had no difference in body weight compared to control animals (**Supplementary Fig. 1f**), they exhibited significantly improved systemic glucose homeostasis (**Supplementary Fig. 1g**), suggesting that hepatic ACOX1 inactivation has a primary effect on insulin sensitivity, independent of body weight difference. Since adipose tissue inflammation affects insulin sensitivity, we assessed inflammation in gWAT, a visceral adipose tissue depot. It is known that adipose tissue inflammation in the context of obesity is more intense in visceral WAT as compared to subcutaneous WAT^{14,15}. Our data show that Acox1-LKO mice have decreased recruitment of proinflammatory macrophages in gWAT (**Fig. 1n-s**). Concomitantly, our results show that insulin-stimulated phosphorylation of Akt is increased in gWAT, suggesting that the decreased inflammation contributes to the improved insulin signaling/sensitivity. These results are likely mediated by accumulation of ω -3 VLCFAs, such as THA, which our results show activates GPR120, a known regulator adipose tissue inflammation and insulin sensitivity.

6. In Figure 2B, the browning effect is only really detectable in chow diet fed mice. This appears clear from ED Figure 2F (displaying the HFD animals), where only a fairly small and variable effect on Ucp1 mRNA expression is observed. This argues again for a very mild browning effect that is almost entirely abrogated by HFD.

Response: As discussed above and shown in **Figure A**, high fat feeding causes a dramatic reduction in the expression brown gene genes in iWAT. Nevertheless, we performed iWAT gene expression in additional HFD-fed mice and combined the result our previous data. New **Supplementary Fig. 2f** shows that browning markers UCP1, Cidea, PPARa, PGC1a, TBX expressions were significantly higher in Acox1-LKO iWAT. We also confirmed the iWAT browning by Western blot analysis of UCP1 protein levels (**Supplementary Fig. 2i**).

Similar to our results, a recently published study (PMID: 38036537) reported adipose browning under chow diet feeding that was diminished with HFD, but still led to significantly decreased body weight in HFD fed mice. Moreover, the magnitude of gene expression changes was comparable to ours. This suggests that adipose tissue browning is a primary effect of hepatic Acox1 inactivation. However, the browning in the setting of chow diet does not lead to decreased body weight because the mice are already lean.

7. In general, the western blots only display protein levels of UCP1 where the fold-changes are rather small and somewhat variable, what about the abundance of other OXPHOS proteins?

Response: Thank you for this suggestion. We have added blots for mitochondrial protein Tomm20 and/or various OXPHOS proteins to revised **Fig. 2d**, **Fig. 6j** and **Supplementary Fig. 6l**. The increased expression of these proteins is consistent with the notion that hepatic Acox1 KO leads to adipose tissue browning and increased mitochondrial biogenesis.

8. The studies in Figure 4 were done using the same concentration (50 mikrom) for EPA, THA and HHA.

How was this concentration chosen? Did the authors perform any titration experiments? Data in Figure 6 suggest that THA levels are around 2 μM in wildtype mice and 15 μM in the *Acox1* KO.

Response: We did perform titration experiments before selecting the dose for fatty acid treatment. Concentrations including 10 μM , 25 μM , 50 μM and 200 μM were tested. The highest concentration (200 μM) of THA and HHA had toxic effects and led to adipocyte death. The effect of 10 μM , 25 μM , 50 μM fatty acid on UCP1 expression is shown in **Figure E** below. The results show that 50 μM THA and HHA had the highest browning effect without toxicity. Thus, we selected 50 μM for further experiments. Even lower doses, such as 10 μM and 25 μM of THA and HHA, showed mild browning effects.

9. In ED Figure 4, how was the expression of white adipocyte marker genes affected? For example: LIPE, PNPLA2, PLIN1 and ADIPOQ.

Response: We added additional markers to **Supplementary Fig. 4b**, including PLIN1, FASN and COX4. Although the browning marker COX4 was increased with THA treatment, PLIN1 and FASN were not affected.

10. The data in Figure 5G show a rather small effect of THA on UCP1 mRNA (40% or so) and protein levels (not quantified). In Figure 4D the effect on Ucp1 mRNA is around 250%. This is still a very low effect as browning usually induces >10-100-fold increases in Ucp1 gene expression in white adipocytes.

Response: The effect of THA on UCP1 expression have been done several times, and we usually see 200%-300% increase of UCP1 expression. We repeated this experiment and obtained consistent results (see revised **Fig. 5g**). The >10-100-fold increase in UCP1 gene expression reported in some publications is likely due to virtually undetectable UCP1 expression under baseline conditions, such as in 3T3-L1 cells. In such

cases, even a small absolute increase seems large in terms of fold change. All our in vitro experiments were done using SVF cells derived from iWAT that were differentiated in medium containing 1nM T3, which promotes UCP1 gene expression. Thus, our cells already express UCP1 prior to THA treatment. Details regarding our adipocyte differentiation procedure are provided in the Methods section (pages 17-18). The revised **Fig. 5h** shows quantification of UCP1 protein levels. Given that mitochondrial respiration (**Fig. 4i**), Akt phosphorylation (**Fig. 4j**) and Glut4 translocation (**Fig. 4k**) were significantly increased, the effects of THA treatment are physiological relevant.

11. The weakest part are the data in Figure 6. Treatment with THA results in small increases in serum-THA (Figure 6B) without any effects on body weight, body composition or glucose tolerance (despite the authors stating that there is an effect, all p-values are >0.05) (Figure 6D-F). In line with this, there are no significant increases in browning gene markers (Figure 6I) except for UCP1 which would be non-significant if correction for multiple comparisons would have been done. I don't think the data in this Figure strengthens their hypothesis.

Response: The lack of body weight difference was due to the fact that this was a short-term experiment with a small cohort of mice, primarily because the fatty acid was extremely expensive (\$4300 for 100 mg, sufficient for 2-week treatment). Moreover, as noted in the manuscript, the exogenously supplied THA does not accumulate to the same level as in the Acox1-LKO mice, likely due to its β -oxidation by the hepatic Acox1 in WT mice. Nevertheless, the levels in iWAT were significantly increased, resulting in iWAT browning and improved systemic glucose homeostasis. To validate these results and improve statistical significance, we purchased additional 100 mg of the fatty acid and treated another cohort of WT mice for two weeks. The revised **Fig. 6** with combined data from the two cohorts shows that although body weight was still not altered (**Fig. 6d**), fat mass was significantly decreased while lean mass was increased (**Fig. 6e**) and glucose tolerance was significantly improved (**Fig. 6f**). Moreover, gene expression (**Fig. 6i**) and Western blot (**Fig. 6j**) analyses showed that browning markers and OXPHOS proteins were increased with THA treatment. Together, these data indicate that THA, the major circulating species of ω -3 VLCFA that accumulates in Acox1-LKO mice, mediates the beneficial effects of the knockout.

Reviewer #3 (Remarks to the Author):

This is a well-conducted study that provides novel insights into the role of liver Acox1 in regulating circulating lipid levels, in particular omega-3 VLCFAs and their impact on metabolic health, in particular on the browning of adipose tissue. In vitro experiments indicate that these protective effects of THA are mediated by GPR120 in adipocytes, but in vivo evidence for this mechanism is lacking. In general, the claims of this study are well supported by data, experiments are technically sound, with appropriate controls, and presented in sufficient detail. However, there are several aspects that require some additional experiments and modifications that need to be done in the manuscript to improve clarity. Once these issues are addressed, I think this manuscript will advance our understanding of the role of peroxisomal fatty acid oxidation in metabolic homeostasis and provide more insights about intricate interorgan metabolic connections. Here are my comments.

Response: Thank you for your careful review of our manuscript. We appreciate your thoughtful and insightful comments, which have markedly improved our paper. Our point-by-point response to your comments are provided below.

Major comments

1. One of my main concerns is the lack of data regarding the specific inhibitory effects of TDYA on Acox1 and peroxisomal beta-oxidation. In other words, how are the authors sure that TDYA does not have off-target effects on other enzymes? At least they should add data of classical VLCFA levels such as C26 levels in mice treated with TDYA. Moreover, there should be a discussion about the appearance of differences in BW gain much later than in Acox1-LKO mice. Why not represent food intake in grams as it is usually done? This representation in % of BW is confusing and not informative. But having said all that, without definite proof for a specific inhibition of ACOX1 by TDYA my recommendation is to withdraw these data from the manuscript, which is still very robust even without these data, but its inclusion may confound future research on peroxisomal b-ox inhibition, until we know about the off-target effects of TDYA.

Response: The specific inhibition TDYA on Acox1 has been reported in previous study by Zeng et. (PMID: 28077576). They checked the inhibitory activities of TDYA on several peroxisomal fatty acid beta oxidation enzymes and found only ACOX1 was inhibited. As a 23-carbon long molecule that resembles a VLCFA, TDYA-CoA is likely not transported into mitochondria. We also tested the peroxisomal FAO inhibition effect of TDYA in AML12 cells by measuring chain shortening of D₃-C22:0, a stable isotope-labeled VLCFA. Our results show that the ratios of D₃-C18:0 to D₃-C22:0 and D₃-C16:0 to D₃-C22:0 were decreased by 50%-60% with TDYA treatment (new **Supplementary Fig. 6b**). Moreover, we show that C24:6 was significantly increased with TDYA treatment in mice (**Supplementary Fig. 6c**). These data support the notion that peroxisomal β -oxidation of VLCFA is inhibited by TDYA. However, DHA was not decreased with TDYA treatment as it was with genetic inactivation of Acox1, suggesting that the pharmacological inhibitor has a more complex effect on VLFA levels. In TDYA-treated AML12 cells and mice or Acox1-LKO mice, we did not observe accumulation of C26:0.

It is possible that significant difference in body weight in TDYA-treated emerges later than in Acox1-LKO mice because TDYA-mediated inhibition of Acox1 activity is partial as opposed to genetic ablation of Acox1. Moreover, the pharmacological inhibitor may need more time to accumulate in tissues and target the enzyme. Since TDYA treatment phenotypically resembles hepatic Acox1 KO in several aspects, including decreased diet-induced obesity, increased thermogenic gene expression and improved glucose homeostasis, it is justifiable to keep the TDYA data in the manuscript.

For food intake result, we now present the data as grams/day (revised **Supplementary Fig. 6g**). Again, no difference was observed between control and TDYA groups.

2. The authors claim that the omega-3 VLCFAs they identified are 'previously uncharacterized VLCFA' However, this statement is misleading, as these fatty acids have been known to be metabolized in peroxisomes for decades, at least for THA. Indeed, patients with peroxisomal disorders frequently present deficits in C22:6 (DHA). It is also known that they accumulate in global Acox1 KO mice. While the specific omega-3 VLCFAs they identified may not have been studied in the context of GPR120 activation and metabolic health, their existence and their connection with peroxisomal metabolism are well established. Therefore, this fact should be acknowledged (PMID 8847474, PMID 11734571 and PMID 11855929). The original reference of global Acox1 KO mice that show extensive changes in the liver should be cited too (PMID 8798738).

Response: By “previously uncharacterized” we meant previously unknown physiological role. This language has been clarified in the revised manuscript. Thank you for your suggestions for references. We have cited these papers and discussed the relationship between DHA and C24:6 fatty acid in the Introduction, as well as citing the original reference of Acox1 global KO mice.

3. Given the importance of peroxisomal metabolism in the liver in this manuscript, it will be a great addition to this paper to evaluate peroxisomal number/proliferation in the livers of Acox1-LKO mice, and their changes with HFD. That could help understand the role of hepatic peroxisomes in diet-induced obesity.

Response: Thank you for the comment. Our unpublished results indicate that inactivation of Acox1 results in increased number of peroxisomes. We are pursuing the underlying molecular mechanism that we plan to publish elsewhere. However, we include the following Western blot analysis (**Figure F**) for the reviewers to show that the peroxisomal markers PEX16 and PMP70 are increased in the livers of Acox1-LKO mice.

We also compared the peroxisome markers in the livers of chow and HFD-fed mice and the results are shown in new **Supplementary Fig. 1b**. The results show that the marker proteins, including PEX5, PEX16, and PMP70 were increased in the liver of HFD-fed mice, suggesting that liver peroxisomes regulate diet-induced obesity and related complications.

4. The characterization of browning of white adipocytes induced by serum of Acox1 LKO should be further expanded besides mRNA levels of browning genes, in order to evaluate whether the concentrations present in the Acox1-LKO serum exert similar effects as the ones seen by THA treatment. Thus, repeating the experiments shown in Figure 4 for THA (or at least some of them) with the serum from Acox1-LKO mice will be very informative.

Response: Thank you for the comment. As per your suggestion, we performed Western blot analysis using serum from Acox1-LKO mice. The new results are presented in **Fig. 2i**. Consistent with the gene expression data, Western blot analysis shows that Acox1-LKO serum treatment results in increased levels of UCP1 and COX4, confirming the browning effect of the KO serum.

5. All the claims related to the role of GPR120 in this liver-adipose axis need to be toned down across the manuscript including the title, since authors do not provide in vivo evidence for this role of GPR120. Their experiments in vitro are sound and relevant but not enough to place GPR120 in this axis that require in

vivo communication between these two organs. GPR120 expression could be measured in vivo (chow and HFD in control and Acox1-LKO to get some clues about changes in this GPCR in vivo). Following up on this last comment, authors assume that the decrease in

circulating fatty acids is caused by changes in FAO, but it could be also caused by changes in lipolysis. Considering that agonists of GPR120 have a profound effect on lipolysis (PMID 33091626), this is an aspect that should be investigated in Acox1-LKO mice (chow and HFD conditions).

Response: Thank you for your suggestion. We changed the title of the manuscript to remove reference to GPR120. As per your suggestion, we measured GPR120 expression in iWAT from chow and HFD fed Acox1-LKO and control mice. Consistent with our results linking the increased circulating ω -3 VLCFAs to GPR120, gene expression of the lipid sensor was significantly increased in chow fed Acox1-LKO mice (**Supplementary Fig. 5d**). However, this effect was masked in HFD-fed mice, since GPR120 expression trends lower with high fat feeding.

To determine if altered circulating lipids reflect changes in lipolysis, we measured lipolysis in iWAT of chow and HFD-fed Acox1-LKO and control mice. As shown in **Figure G** below, lipolysis was not affected by the genotype.

Figure G. Measurement of lipolysis in iWAT of control and Acox1-LKO fed chow or HFD for 15 weeks.

6. The use of unpaired t-test to evaluate differences in multiple figures (Figure 4, Figure 6, Ext. Data Fig. 1A-C, and more) does not seem correct as more than two groups are being compared. Please revise the statistical analysis across the manuscript and use the appropriate statistical test to evaluate those differences.

Response: Thank you for your suggestion. We have revised the statistical analysis. Details about statistical tests are provided in the figure legends.

Minor comments

1. In Fig.1, at what point were peroxisomal enzymes expression measured? Only mention in the methods of duration of HFD is 9 weeks before THA treatment, but Fig. 1C shows 11 weeks of HFD treatment. It is

not clear which time point was the one used to measure mRNA. Please clarify. Did authors measure the expression of Acaa1a or Acaa1b? Why n=3 only for gene expression, while in Fig. 1S the n is much higher?

Response: The samples used in **Fig. 1b** are from mice fed for 16 weeks with chow diet or a HFD. The iWAT and BAT mRNA expression analyses were performed in mice after 17 weeks of HFD feeding. We have added the HFD feeding time in figure legends. Acaa1a was measured and this is now specified in Fig. 1b. Samples used in Fig.1b are from WT C57 mice fed with chow or HFD. Since the liver gene expression was relatively stable, N=3 was sufficient to show significant difference between the groups. The mice used in the original **Fig.1S** (now **Supplementary Fig. 1p**) are Acox1-LKO or control mice Acox1^{flox/flox} background. The latter results are combined data from separate cohorts of mice that yielded similar results.

2. The way that the changes in leukocytes in SVF are shown in Ext. Data 1E is confusing. In the text the authors talk about total numbers of SVF cells but they represent per g of fat. Perhaps it would be more visual to show the total number of cells, so it would be clear that there are no differences despite the increase in gWAT weight. Alternatively, the text could be changed to better reflect what has been plotted.

Response: As discussed above in response to Reviewer 1's comment 4, we have repeated the experiment with a larger cohort of mice and obtained consistent results with respect to changes in immune cells. **Supplementary Fig. 1** and main **Fig.1** have been updated with the new data and the associated text has been extensively revised to better reflect the results.

3. EE data. It is not clear in the text whether EE was measured in HFD-fed control and Acox1-LKO mice or in chow-fed. Assuming it was done in HFD-fed mice, body masses are different between control and Acox1-LKO mice. Thus, a regression-based analysis of EE using body mass as a covariate should be included in the analysis of EE. Data looks convincing but that additional analysis would give more information. Indeed authors did this in Ext Data Fig. 6 with other conditions, so why not doing the same here?

Response: We appreciate the suggestion. We have included regression-based analysis of EE in the new **Fig. 2b**.

In that same data from metabolic chambers in Ext. Data Fig. 2, 5 g of HFD per day seems quite high for B6 mice. Do authors have food intake data (ideally cumulative) from the same mice shown in Fig.1, where progression of BW is shown?

Response: The original **Supplementary Fig. 2a** included food intake data from chow-fed animals. The figure has been replaced with data from HFD-fed animals.

4. Fig 2D shows UCP1 IHC in Control and Acox1-LKO. In how many mice was this experiment performed? Information is not available in figure legend. Why was mitochondrial respiration performed in frozen tissue when this is suboptimal in comparison with fresh tissue? Did authors observe more mitochondrial biogenesis in iWAT of Acox1-LKO mice?

Response: IHC (now **Fig. 2e**) was done in two mice for each group. This information is now specified in the figure legend. Assessment of mitochondrial respiration in fresh samples is technically challenging and yields inconsistent results. The assay for frozen samples was recently developed by laboratory of Orian

Shirihai, a leader in the field (PMID: 32432379) and makes measurement of mitochondrial respiration in multiple samples convenient and yields highly consistent results. Our results indicate that various mitochondrial markers were increased in iWAT of *Acox1*-LKO mice (**Fig. 2d**), indicating that mitochondrial biogenesis is increased.

5. In the reporting of changes in genes involved in omega-oxidation it would be useful for the reader to have references to a review discussing the omega-oxidation pathway (either PMID 37140888 or PMID 24126285).

Response: Thank you for this suggestion. PMID 37140888 is cited in the ω -oxidation part of this paper.

6. Add units for scale in Fig. 3B (z-score I assume).

Response: The units “z-score” was added to the figure.

7. If authors have acylcarnitine data from the serum/hepatic lipidome I suggest they include them in a separate graph/s, as it would help to understand the impact of FAO in these *Acox1*-LKO mice, and more importantly since these metabolites have been shown to have a thermogenic role.

Response: In new **Supplementary Fig. 3g**, we report changes in acylcarnitines. Our lipidomic analysis showed a few species of acylcarnitine were increased while others were decreased in *Acox1*-LKO mouse serum (**Supplementary Fig. 3g**). Given that the altered acylcarnitine species were present in low abundance, it is unlikely that the iWAT browning in *Acox1*-LKO mice is mediated by circulating acylcarnitines.

8. Quantification is missing for WB in Fig. 5H

Response: The quantification has been added to **Fig. 5H**.

9. Figure 1A depicts peroxisomal beta-oxidation pathway with the peroxisomal transporter ABCD1. Considering the focus on C24:6n-3 (THA), ABCD2 should be added to that schema, as it is the hypothesized transporter for THA, and there it no mention to it across the manuscript (PMID 21145416 and PMID 18854420).

Response: ABCD1 has been changed to ABCD1/2 in **Figure1A** and added the following text:

Very long chain fatty acyl-CoA are transported into peroxisomes for β -oxidation by ABCD1 or ABCD2, which have distinct substrate specificities¹¹

- Other comments:

- Intro: Following up on my previous comments regarding omega-3 VLCFA and peroxisomes, authors should change the introduction to acknowledge this previous knowledge, and also include branched-chain fatty acids in the substrates metabolized by peroxisomes. I would also suggest adding to the intro that VLCFA beta-oxidation is severely reduced in the liver of ACOX1-LKO mice, so the reader is aware of that before diving into the results where it is mentioned that peroxisomal beta-ox is dysfunctional in ACOX1-LKO.

Response: Thank you for this suggestion. As noted in response to Reviewer 1's comment 2 and 3, we edited the Introduction to expand the discussion on substrate specificities of different ACOX family members and mention the role of ACOX1 in β -oxidation of ω -3 VLCFA by Acox1 and the role of ACOX2 in β -oxidation of branched-chain fatty acids. The information that Acox1 KO leads to severely impaired β -oxidation of VLCFA was also added to the Introduction.

- Methods/Reporting:

Please provide more information about the analysis of the metabolome with Metaboanalyst to improve reproducibility of the findings. How were the metabolites imputed in Metaboanalyst (KEGG ids or HMDB ids, and how many were used) to perform the pathway enrichment?

Response: The following detail regarding pathway analysis was added to the Methods: "The pathway enrichment analysis was performed in MetaboAnalyst through input of the compound names of 313 changed lipids which matched to the MetaboAnalyst database."

How were TG measured?

Response: The Methods section was revised to include a description of serum TAG assay: "Serum TAG levels were measured using the Triglycerides Reagent (Thermo Scientific, #TR22421) and Triglyceride Standard (Pointe Scientific, #T7531STD). In brief, 2.5 μ l of the standard or serum sample and 250 μ l of the reagent were added to each microplate well. After incubating at 37 $^{\circ}$ C for 10 minutes, absorbance at 500 nm was measured using a Tecan plate reader."

The method for measurement of fecal TG is also described.

Please report the total number of animals used for this study. There is no mention regarding the sex of the animals used in this study.

Response: The number of animals used for each experiment are indicated in figure legends. Most of the experiments were done with male animals and selected experiments metabolic phenotyping experiments were done using female mice.

The inclusion of the tissue used for measurements in the figure panels would facilitate the comprehension of these large figures.

Response: Tissue information has been added to Figures.

Please provide details regarding the antibodies used in this study and full blots of Western Blots with MW markers.

Response: We have added MW markers to the Western blot panels. The original uncut blots are provided in the source data file. The information on antibodies is provided in the Reporting Summary.

- Writing: Make a consistent use of beta/ β atin and greek nomenclature. p6, L174-175: ... circumstances that resulting in the... (result?)

Thank you for your suggestion. “Beta” has been changed to “ β ”; “omega” has been changed to “ ω ”. We have corrected the typo.

- Discussion:

Authors claim that “Whether peroxisomal β -oxidation influences systemic metabolism to affect diet-induced obesity has not been investigated”. That is an incorrect statement. Zeng et al. did that using TDYA, the inhibitor authors used in this manuscript. Therefore, this sentence should be changed to reflect a more specific claim about the authors’ findings.

Response: We changed this sentence to “Whether **liver** peroxisomal β -oxidation influences systemic metabolism to affect diet-induced obesity has not been investigated.” to highlight the role of peroxisomal β -oxidation in the liver tissue.

Acox1 inactivation and hepatomegaly. I think this is an important aspect that has not been properly discussed. What is the hypothesis of the authors for this discrepancy between global KO (even Acox1Lampe) and Acox1-LKO mice, even more when RNA-seq data show activation of PPAR signaling pathway and multiple PPAR-alpha targets, that in mice drive hepatomegaly.

Response: We discussed the discrepancy in our response to Reviewer 2’s comment #1 and on page 15 of the revised manuscript. In addition, as we reported previously (He et al. 2020; PMID:32473093), Acox1-LKO mice are protected against hepatic steatosis through induction of lipophagy. Mice with global KO of Acox1 on 129/Sv background, which is known for having increased susceptibility for developing cancer, exhibit steatohepatitis and develop hepatic carcinomas in the context of increased endoplasmic reticulum (ER) stress as they age (Huang et al., 2011; PMID: 21801867). Moreover, Acox1Lampe1 mice (also on 129/Sv background), exhibit hepatosteatorosis that progresses to steatohepatitis and ultimately hepatocellular carcinoma (Sheridan et al., 2011; PMID: 21760938). Bone marrow transplantation from Acox1Lampe1 mice into wild-type recipients recapitulated the hepatocellular damage and systemic inflammation, suggesting that Acox1 inhibition in immune cells, rather than hepatocytes themselves, contributes to the phenotype (Moreno-Fernandez et al., 2018; PMID: 29563328).

REVIEWERS' COMMENTS

Reviewer #1 (Remarks to the Author):

I think the manuscript has improved significantly and all questions and open points could be clarified. I have no further comments.

Reviewer #2 (Remarks to the Author):

No further comments.

Reviewer #3 (Remarks to the Author):

The authors should be commended for their immense effort to improve this manuscript, which now presents more robust and nuanced conclusions. This revised version undoubtedly represents an important contribution to our understanding of the roles of peroxisomes in metabolic homeostasis.

I have just a few minor comments regarding this revised version:

- In the first paragraph of the discussion, the authors conclude: "Liver-specific knockout of the peroxisomal β -oxidation enzyme ACOX1 promotes resistance to diet-induced obesity...". I suggest specifying that this is only true in male mice (no difference in BW gain in female mice under HFD). It would be helpful to specify this limitation and possibly discuss the differences in body weight gain between Acox1-LKO mice fed a high-fat diet and wild-type mice fed a chow diet (data already present in the manuscript in different figures). This comparison could provide a clearer picture of the effect size in protecting against diet-induced obesity.
- While the addition of VLCFA oxidation measurements with TDYA using labeled peroxisomal substrates is valuable, some questions remain regarding the compound's specific action. Notably, the livers of mice treated with TDYA do not show the decrease in C22:6 (DHA) levels observed in Acox1-LKO livers. There is no data in the manuscript regarding the levels of THA in serum after the treatment, leaving open the mechanism of action of TDYA on changes in metabolic balance. Moreover, the effect size of TDYA in the protection against diet-induced obesity is rather small, appearing after many weeks of HFD. The authors discuss the possible causes of this late phenotype in the response to reviewers but not in the manuscript. I think these are relevant topics to add to the discussion of the manuscript. This will provide readers with a more comprehensive understanding of the findings and a more clear picture of the interpretation of the results obtained with this inhibitor.

RE: Manuscript No. NCOMMS-23-39474A

“Liver ACOX1 Regulates Levels of Circulating Lipids that Promote Metabolic Health through Adipose Remodeling” by Lu et al.

REVIEWERS' COMMENTS

Reviewer #1 (Remarks to the Author):

I think the manuscript has improved significantly and all questions and open points could be clarified. I have no further comments.

Response: Thank you for your help in improving our paper.

Reviewer #2 (Remarks to the Author):

No further comments.

Reviewer #3 (Remarks to the Author):

The authors should be commended for their immense effort to improve this manuscript, which now presents more robust and nuanced conclusions. This revised version undoubtedly represents an important contribution to our understanding of the roles of peroxisomes in metabolic homeostasis.

Response: Thank you for commending our efforts. We appreciate your thoughtful comments. Our point-by-point responses to your comments are provided below.

I have just a few minor comments regarding this revised version:

- In the first paragraph of the discussion, the authors conclude: “Liver-specific knockout of the peroxisomal β -oxidation enzyme ACOX1 promotes resistance to diet-induced obesity...”. I suggest specifying that this is only true in male mice (no difference in BW gain in female mice under HFD). It would be helpful to specify this limitation and possibly discuss the differences in body weight gain between Acox1-LKO mice fed a high-fat diet and wild-type mice fed a chow diet (data already present in the manuscript in different figures). This comparison could provide a clearer picture of the effect size in protecting against diet-induced obesity.

Response: Thank you for the suggestion. We revised the Discussion (p13) to “Liver-specific knockout of the peroxisomal β -oxidation enzyme ACOX1 promotes resistance to diet-induced obesity, decreases infiltration of pro-inflammatory macrophages into adipose tissue **in male mice**, and improves systemic glucose homeostasis **in both male and female mice.**” to specify the limitation of phenotypes in male and female mice.

- While the addition of VLCFA oxidation measurements with TDYA using labeled peroxisomal substrates is valuable, some questions remain regarding the compound's specific action. Notably, the livers of mice treated with TDYA do not show the decrease in C22:6 (DHA) levels observed in Acox1-LKO livers. There is no data in the manuscript regarding the levels of THA in serum after the treatment, leaving open the mechanism of action of TDYA on changes in metabolic balance. Moreover, the effect size of TDYA in the protection against diet-induced obesity is rather small, appearing after many weeks of HFD. The authors discuss the possible causes of this late phenotype in the response to reviewers but not in the manuscript. I think these are relevant topics to add to the discussion of the manuscript. This will provide readers with a more comprehensive understanding of the findings and a more clear picture of the interpretation of the results obtained with this inhibitor.

Response: Thank you for your suggestion. We added the following text on page 12 of the revised manuscript:

“However, the effects of genetic and pharmacological inhibition of ACOX1 were not identical. For example, the body weight difference in TDYA-treated mice is smaller and emerges later as compared to *Acox1*-LKO animals. This is presumably because TDYA-mediated inhibition of ACOX1 activity is partial and not liver-specific as opposed to genetic ablation of *Acox1*. Moreover, the pharmacological inhibitor may need more time to accumulate in tissues and target the enzyme. Nevertheless, the two models resemble in several aspects, including decreased diet-induced obesity, increased thermogenic gene expression and improved glucose homeostasis.”